# Towards a Unified Theory of Quantization and Sparsity

## Abstract

Quantization and sparsification are two model compression strategies that are traditionally treated as orthogonal in the literature. Building on recent work, we show that jointly considering these techniques can meaningfully affect compression performance. First, we extend prior tensor-level analyses and prove that for any $L_p$ norm, applying sparsification before quantization ($\mathbf{S} \to \mathbf{Q}$) always yields lower errors than the reverse. However, we demonstrate that tensor-level analysis is insufficient to predict model performance, motivating the need for model-level evaluation. As such, we provide the first model-level analysis showing that $\mathbf{S} \to \mathbf{Q}$ obtains better loss in certain settings when we choose quantization and sparsification algorithms independently. Yet, this preference does have its limits. When fully relaxing model assumptions, we find it difficult to prove the superiority of $\mathbf{S} \to \mathbf{Q}$, casting doubt on the preference in the general case. To that end, we introduce Quantization-Aware Sparsification (QAS), a novel compression framework that sparsifies accounting for prior quantization, as a simple counterexample. Using this framework, we provide a simple counterexample in which $\mathbf{Q} \to \mathbf{S}$ using QAS performs comparably to $\mathbf{S} \to \mathbf{Q}$, illustrating that careful co-design between model compression steps can greatly influence performance.

## 1 Introduction

Given the growing demand to deploy machine learning models on resource-constrained devices, model compression techniques have become increasingly important. Two of the most widely adopted methods are quantization and sparsification, both of which reduce memory and computational overhead, making them well-suited for deployment on edge devices with strict energy and memory limitations.

Quantization compresses models by reducing the bitwidth of weights and activations, typically converting 32-bit floating-point values into lower-precision formats such as 8-bit integers (Dettmers et al., 2022). In this paper, we focus on post-training quantization, where models are first trained in full-precision and then quantized for inference (Frantar et al., 2023a; Lin et al., 2024). This approach simplifies deployment but can introduce non-negligible accuracy degradation if not handled carefully.

Sparsification, in contrast, reduces model size by setting a fraction $p\%$ of the weights to zero. These zero values can be exploited during inference to skip computations entirely, leading to both speed and energy savings. Unlike quantization, which affects all parameters uniformly, sparsification introduces structural changes into the model, potentially disrupting its learned representations.

Although both techniques are widely studied, they are typically treated independently. Quantization and sparsification methods are proposed independently. However, it is unclear that applying quantization and sparsification sequentially will compound their benefits. In fact, the order of application for each compression step can significantly influence final model accuracy (Harma et al., 2025).

The focus of this work will be on the setting where no retraining is done. Retraining is costly, and we ideally want to implement these methods where minimal to no retraining is required. Furthermore, if retraining is truly desired, we can use change in loss as a proxy for amount of retraining necessary.

In this paper, we will make the following contributions to the literature.

- We extend the current analysis of the ordering of quantization and sparsification at the tensor level. In particular, we show that sparsification then quantization ($\mathbf{S} \to \mathbf{Q}$) leads to smaller tensor-level error than quantization then sparsification ($\mathbf{Q} \to \mathbf{S}$) for arbitrary $L_p$ norm with $p \geq 1$ in the naive setting.

- We demonstrate the fact that tensor-level metrics are insufficient to predict model behavior under compression. This motivates the need for holistic, model-level evaluation.

- Using model-level evaluation, we show that $\mathbf{S} \to \mathbf{Q}$ is preferred when pairing the naive tensor-wise quantization scheme with tensor-wise magnitude-based or Optimal Brain Damage (OBD) sparsification schemes, assuming the Hessian of the loss function is diagonal. To the best of our knowledge, this is the first result relating the order of sparsification and quantization to the model loss.

- We show that $\mathbf{S} \to \mathbf{Q}$ may not always be preferred in the general setting though. We introduce **Quantization-Aware Sparsification (QAS)**, a unified compression framework that attempts to remedy the $\mathbf{Q} \to \mathbf{S}$ direction. QAS is designed to show that the appropriate choice of algorithms can yield minimal performance degradation, regardless of the choice of ordering.

The rest of the paper is structured as follows. In Section 2, we discuss the current literature surrounding the intersection of quantization and sparsification. In Section 3, we present a mathematical formulation of the two problems. In Section 4, we present our tensor-level results regarding the order of quantization and sparsification. In Section 5, we extend this analysis to the model level. In Section 6, we introduce QAS, our proposed sparsification scheme that demonstrates that $\mathbf{Q} \to \mathbf{S}$ need not be suboptimal.

## 2 RELATED WORKS

Harma et al. (2025) is one of the first works to systematically examine the interaction between quantization and sparsification. Notably, they argue that these two compression techniques should not be treated as orthogonal, analyzing their interplay at both the tensor and dot-product levels. They demonstrate that, at both levels, the total error resulting from the combined application of quantization and sparsification—regardless of order—can exceed the sum of the individual errors. Additionally, they show that at the tensor level, the error incurred by applying $\mathbf{S} \to \mathbf{Q}$ is less than or equal to the error from applying $\mathbf{Q} \to \mathbf{S}$, under the $L_1$ norm and assuming a naive symmetric max-scaled quantization scheme and magnitude-based sparsification scheme.

However, their theoretical analysis is limited in that it focuses exclusively on naive, max-scaled tensor-wise quantization schemes and magnitude-based sparsification. Moreover, the analysis analyzes the effect on the tensor level but does not consider the effect of these compression methods on the model's loss, making it difficult to extrapolate the results to overall model accuracy. In this work, we build upon the contributions of Harma et al. (2025) by generalizing the tensor-level comparison of $\mathbf{Q} \to \mathbf{S}$ and $\mathbf{S} \to \mathbf{Q}$ to general $L_p$ norms, by performing analysis at the model level, and by introducing QAS, a compression framework that can recover performance degradation under $\mathbf{Q} \to \mathbf{S}$.

## 3 PROBLEM FORMULATION

### 3.1 QUANTIZATION

A quantization scheme $\left( \Phi, \{\mathcal{Q}_i, Q_i, D_i\}_{i=1}^d \right)$ consists of a parameter function $\Phi : \mathbb{R}^d \to \Theta$, a set of tuples containing quantization levels $\mathcal{Q}_i \subseteq \mathbb{R}$, a quantization function $Q_i : \mathbb{R}^d \times \Theta \to \mathcal{Q}_i$, and a dequantization function $D_i : \mathcal{Q}_i \times \Theta \to \mathbb{R}$ for all $i \in [d]$. Intuitively, we have that $\Phi$ calculates metrics that can be used to compute quantization parameters like scale. Then, $Q_i$ quantizes the weight $\mathbf{w}_i$ to some $\mathbf{q}_i \in \mathcal{Q}_i$. Then, $D_i$ dequantizes $\mathbf{q}_i$, using the quantization parameters, to some value in $\mathbb{R}$. We require that $|\mathcal{Q}_i| = 2^{b_i}$, where $b_i$ is the desired bitwidth for index $i$. Note that there is a quantization and dequantization for every index, since the scales could differ for different elements. We can add different constraints depending on the granularity of quantization, which we discuss in Section A.

Now, we build up towards a unifying quantization function that can be applied to $\mathbf{w} \in \mathbb{R}^d$. First, let $\mathbf{Q} : \mathbb{R}^d \times \Theta \to \prod_{i=1}^d \mathcal{Q}_i$ be defined by $(\mathbf{Q}(\mathbf{w}, \theta))_i = Q_i(\mathbf{w}, \theta)$ for all $\mathbf{w} \in \mathbb{R}^d$, $\theta \in \Theta$, $i \in [d]$. Intuitively, $\mathbf{Q}$ applies $Q_i$ to the element at index $i$. Similarly, define $\mathbf{D} : \prod_{i=1}^d \mathcal{Q}_i \times \Theta \to \mathbb{R}^d$ by applying $D_i$ element-wise, so that $(\mathbf{D}(\mathbf{q}, \theta))_i = D_i(\mathbf{q}_i, \theta)$ for all $\mathbf{q} \in \prod_{i=1}^d \mathcal{Q}_i$, $\theta \in \Theta$, $i \in [d]$.

Then, to quantize $\mathbf{w}$, we define $\overline{\mathbf{Q}} : \mathbb{R}^d \to \mathbb{R}^d$ by the map

$$\mathbf{w} \mapsto \mathbf{D}(\mathbf{Q}(\mathbf{w}, \Phi(\mathbf{w})), \Phi(\mathbf{w})).$$

We have that $\overline{\mathbf{Q}}$ maps weights to a quantization grid that still lives in $\mathbb{R}^n$.

## 3.2 Sparsity

A sparsification scheme $\{S_i\}_{i=1}^d$ consists of a set of pruning functions $S_i : \mathbb{R}^d \to \{0, 1\}$ for each weight $\mathbf{w}_i$. Intuitively, if $S_i(\mathbf{w})$ maps to 0, then we will prune $\mathbf{w}_i$; else, we will retain $\mathbf{w}_i$. Then, to achieve $p\%$ sparsity, we enforce the requirement that for all $\mathbf{w} \in \mathbb{R}^d$, we have that

$$|\{S_i(\mathbf{w}) = 0 \mid i \in [d]\}| \geq p\% \cdot d.$$

This enforces that for any weight input, at least $p\%$ of the weights should be pruned. Thus, we define $\overline{\mathbf{S}} : \mathbb{R}^d \to \mathbb{R}^d$ such that

$$\overline{\mathbf{S}}(\mathbf{w})_i = \mathbf{w}_i \cdot S_i(\mathbf{w}).$$

This provides a general framework for choosing weights to prune for a model.

## 3.3 Objective

As done in prior work, the change in loss from quantization and sparsification can be approximated using a second-order approximation (Nagel et al., 2020; Li et al., 2021; LeCun et al., 1989; Hassibi & Stork, 1992). In particular, they view quantization and sparsification as minor perturbation of the weights. Using a second-order Taylor approximation implies that

$$L(\hat{\mathbf{w}}) - L(\mathbf{w}) = \frac{1}{2}\Delta\mathbf{w}^\mathsf{T}\mathbf{H}_L(\mathbf{w})\Delta\mathbf{w} + \mathcal{O}\left(\|\Delta\mathbf{w}\|^3\right)$$

where $\hat{\mathbf{w}}$ is the perturbed weights, $L$ is the loss function, and $\Delta\mathbf{w} = \hat{\mathbf{w}} - \mathbf{w}$. For the rest of the paper, let $h_{ii} = \frac{\partial^2 L}{\partial \mathbf{w}_i^2}$ denote the $i$th diagonal element of the Hessian matrix $\mathbf{H}_L(\mathbf{w})$.

## 4 Tensor-Level Analysis

Prior work has mainly focused on naive symmetric max-scaled tensor-wise quantization schemes and magnitude-based tensor-wise sparsification schemes. We mathematically define these in Appendix A and B, respectively. Using this formalism, we now present the main theorem of this section. For a single tensor, we will show that $\mathbf{S} \to \mathbf{Q}$ is preferred for arbitrary $L_p$ norm metric for a naive symmetric max-scaled tensor-wise quantization scheme and magnitude-based tensor-wise sparsification scheme.

**Theorem 1.** *Let $\overline{\mathbf{Q}}$ be a naive symmetric tensor-wise max-scaled quantization scheme and $\overline{\mathbf{S}}$ be the magnitude-based tensor-wise sparsification scheme. Furthermore, assume that $0 \in \mathcal{Q}_i$ for $i \in [d]$. Then, we claim that*

$$\left\|\mathbf{w} - \overline{\mathbf{Q}}(\overline{\mathbf{S}}(\mathbf{w}))\right\|_p \leq \left\|\mathbf{w} - \overline{\mathbf{S}}(\overline{\mathbf{Q}}(\mathbf{w}))\right\|_p$$

*for all $\mathbf{w} \in \mathbb{R}^d$ and $p \in [1, \infty]$.*

Harma et al. (2025) proves the above statement for $p = 1$. In this work, we prove the more general statement for $p \geq 1$. This statement is stronger than the $L_1$ case, since the inequality can have different use cases depending on the choice of $p$. The full proof of this statement can be found in Appendix C. In this section, we provide a quick sketch and explain the implications of the result.

### 4.1 PROOF SKETCH

We prove the statement by proving

$$\sum_{i \in T} \left| \mathbf{w}_i - \overline{\mathbf{Q}}\left(\overline{\mathbf{S}}\left(\mathbf{w}\right)\right)_i \right|^p \leq \sum_{i \in T} \left| \mathbf{w}_i - \overline{\mathbf{S}}\left(\overline{\mathbf{Q}}\left(\mathbf{w}\right)\right)_i \right|^p$$

inductively on the size of set $T \subseteq [d]$. The key insight of the proof is that $\left| \mathbf{w}_i - \overline{\mathbf{S}}\left(\overline{\mathbf{Q}}\left(\mathbf{w}\right)\right)_i \right| = \left| \mathbf{w}_i - \overline{\mathbf{Q}}\left(\overline{\mathbf{S}}\left(\mathbf{w}\right)\right)_i \right|$ if $\mathbf{w}_i$ is pruned and $\left| \mathbf{w}_i - \overline{\mathbf{Q}}\left(\overline{\mathbf{S}}\left(\mathbf{w}\right)\right)_i \right| = \left| \mathbf{w}_i - \overline{\mathbf{Q}}\left(\mathbf{w}\right)_i \right|$ otherwise. This follows from the naive and max-scaled assumption, since the scale remains the same regardless of the pruning.

The main intuition of the proof is that swapping the orders can only create a difference when $\mathbf{w}_i$ is initially pruned and $\mathbf{w}_j$ is not, but the roles are reversed after quantization. Mathematically, we initially have $|\mathbf{w}_i| \leq |\mathbf{w}_j|$. By the order-preserving property, $\left| \overline{\mathbf{Q}}\left(\mathbf{w}_i\right) \right| \leq \left| \overline{\mathbf{Q}}\left(\mathbf{w}_j\right) \right|$. Thus, the roles can only be reversed if $\left| \overline{\mathbf{Q}}\left(\mathbf{w}_i\right) \right| = \left| \overline{\mathbf{Q}}\left(\mathbf{w}_j\right) \right|$. We call this event a *collision*. Applying quantization first leads to these collisions, where previously distinct weights collide and become indistinguishable. In other words, sparsification has a hard time deciding the correct weights to prune if quantization has already occurred.

### 4.2 IMPLICATIONS

Theorem 1 states that for all $p \in [1, \infty]$,

$$\left\| \mathbf{w} - \overline{\mathbf{Q}}\left(\overline{\mathbf{S}}\left(\mathbf{w}\right)\right) \right\|_p \leq \left\| \mathbf{w} - \overline{\mathbf{S}}\left(\overline{\mathbf{Q}}\left(\mathbf{w}\right)\right) \right\|_p .$$

Thus, we have shown that for all $L_p$ error metrics, $\mathbf{S} \to \mathbf{Q}$ is at least as good as $\mathbf{Q} \to \mathbf{S}$. One implication is that

$$\left\| \mathbf{w} - \overline{\mathbf{Q}}\left(\overline{\mathbf{S}}\left(\mathbf{w}\right)\right) \right\|_\infty \leq \left\| \mathbf{w} - \overline{\mathbf{S}}\left(\overline{\mathbf{Q}}\left(\mathbf{w}\right)\right) \right\|_\infty ,$$

which tells us the worst-case error inside the tensor is better for $\mathbf{S} \to \mathbf{Q}$ compared to $\mathbf{Q} \to \mathbf{S}$. In Section 5, we will also see an application of the $L_2$ case of Theorem 1 for analyzing the change in the loss function. For naive quantization schemes and the magnitude-based sparsification scheme, Theorem 1 implies that $\mathbf{S} \to \mathbf{Q}$ is preferred at the tensor level.

### 4.3 NECESSARY CONDITIONS

It is important to note that the assumption of a naive symmetric max-scaled tensor-wise quantization scheme is necessary. We provide a full counterexample in Appendix D that uses AdaRound (Nagel et al., 2020). We replace the naive $Q_i$ with a $Q_i$ that arbitrarily chooses to round up or down. Intuitively, by removing the naive assumption, we lose guarantees about our quantization scheme at the tensor level. We no longer know whether weights will be rounded to the nearest point in the quantization grid. However, removing some guarantees at the tensor level can enable better performance at the model level, where model-level performance is quantified by the objective introduced in Section 3.3.

### 4.4 LIMITATIONS

Although we have a strong result at the tensor level, the change in model performance at the model level is of more interest. In particular, we want to know how the order of sparsification and quantization affects the loss for the entire model, rather than how it affects the error in just the tensor of weights. We can try to extend the tensor-level analysis to the model level using Theorem 1, but it has limitations.

Consider a loss function $L$ that is Lipschitz continuous. Then, we have that

$$\left\| L\left(\mathbf{a}\right) - L\left(\mathbf{b}\right) \right\|_p \leq M_p \left\| \mathbf{a} - \mathbf{b} \right\|_p ,$$

for $\mathbf{a}, \mathbf{b} \in \mathbb{R}^d$ and $M_p \in \mathbb{R}_{>0}$. Thus, we can show that

$$\left\| L\left(\mathbf{w}\right) - L\left(\overline{\mathbf{Q}}\left(\overline{\mathbf{S}}\left(\mathbf{w}\right)\right)\right) \right\|_p \leq M_p \left\| \mathbf{w} - \overline{\mathbf{Q}}\left(\overline{\mathbf{S}}\left(\mathbf{w}\right)\right) \right\|_p$$

$$\left\| L\left(\mathbf{w}\right) - L\left(\overline{\mathbf{S}}\left(\overline{\mathbf{Q}}\left(\mathbf{w}\right)\right)\right) \right\|_p \leq M_p \left\| \mathbf{w} - \overline{\mathbf{S}}\left(\overline{\mathbf{Q}}\left(\mathbf{w}\right)\right) \right\|_p .$$

Theorem 1 gives us that $M_p \left\| \mathbf{w} - \overline{\mathbf{Q}} \left( \overline{\mathbf{S}} \left( \mathbf{w} \right) \right) \right\|_p \leq M_p \left\| \mathbf{w} - \overline{\mathbf{S}} \left( \overline{\mathbf{Q}} \left( \mathbf{w} \right) \right) \right\|_p$. Therefore, we can have a better error bound on Lipschitz functions of $\overline{\mathbf{Q}} \left( \overline{\mathbf{S}} \left( \mathbf{w} \right) \right)$ compared to $\overline{\mathbf{S}} \left( \overline{\mathbf{Q}} \left( \mathbf{w} \right) \right)$. However, without additional assumptions, we cannot guarantee that

$$\left\| L \left( \mathbf{w} \right) - L \left( \overline{\mathbf{Q}} \left( \overline{\mathbf{S}} \left( \mathbf{w} \right) \right) \right) \right\|_p \leq \left\| L \left( \mathbf{w} \right) - L \left( \overline{\mathbf{S}} \left( \overline{\mathbf{Q}} \left( \mathbf{w} \right) \right) \right) \right\|_p.$$

Intuitively, we expect that a feature's importance is not necessarily captured in its magnitude. Thus, the tensor-level results in this section may not necessarily translate to model-level results. Ideally, we would like to make such comparisons, as most practical applications are more concerned with effects on model performance rather than at the tensor level. This motivates the need for model-level analysis, which we present in the next section.

## 5 MODEL-LEVEL ANALYSIS

A key contribution of this work is to generalize the ideas from the tensor level to the model level. Working at the model level is more ideal, since the objective itself, as defined in Section 3.3, is at the model level.

Let $\mathbf{w} \in \mathbb{R}^d$ be the trained weights with respect to loss function $L : \mathbb{R}^d \to \mathbb{R}$. Suppose we have a quantization scheme $\overline{\mathbf{Q}}$ and a sparsification scheme $\overline{\mathbf{S}}$. Then, let $\Delta \mathbf{w_Q} = \overline{\mathbf{Q}} \left( \mathbf{w} \right) - \mathbf{w}$ and $\Delta \mathbf{w_S} = \overline{\mathbf{S}} \left( \mathbf{w} \right) - \mathbf{w}$ be the quantization and sparsification error on the original trained weights, respectively. Let $\Delta \mathbf{w_{S|Q}} = \overline{\mathbf{Q}} \left( \overline{\mathbf{S}} \left( \mathbf{w} \right) \right) - \overline{\mathbf{S}} \left( \mathbf{w} \right)$ be the additional quantization error on sparsified weights and $\Delta \mathbf{w_{Q|S}} = \overline{\mathbf{S}} \left( \overline{\mathbf{Q}} \left( \mathbf{w} \right) \right) - \overline{\mathbf{Q}} \left( \mathbf{w} \right)$ be the additional sparsification error on quantized weights. Finally, let $\Delta \mathbf{w_{Q \to S}} = \overline{\mathbf{Q}} \left( \overline{\mathbf{S}} \left( \mathbf{w} \right) \right) - \mathbf{w}$ and $\Delta \mathbf{w_{S \to Q}} = \overline{\mathbf{S}} \left( \overline{\mathbf{Q}} \left( \mathbf{w} \right) \right) - \mathbf{w}$ be the errors of applying $\mathbf{S} \to \mathbf{Q}$ and $\mathbf{Q} \to \mathbf{S}$, respectively.

After applying $\mathbf{Q} \to \mathbf{S}$, we get weights $\hat{\mathbf{w}}_{\mathbf{Q \to S}} = \mathbf{w} + \Delta \mathbf{w_{Q \to S}}$. Similarly, after applying $\mathbf{S} \to \mathbf{Q}$, we get weights $\hat{\mathbf{w}}_{\mathbf{S \to Q}} = \mathbf{w} + \Delta \mathbf{w_{S \to Q}}$. The quantity of interest is $L \left( \hat{\mathbf{w}}_{\mathbf{Q \to S}} \right) - L \left( \hat{\mathbf{w}}_{\mathbf{S \to Q}} \right)$, the difference in loss between the loss from $\mathbf{Q} \to \mathbf{S}$ and $\mathbf{S} \to \mathbf{Q}$. Intuitively, we would hope for this difference to exhibit a consistent sign, which would imply that one ordering universally outperforms the other.

### 5.1 NAIVE SYMMETRIC TENSOR-WISE MAX-SCALED QUANTIZATION SCHEME AND TENSOR-WISE MAGNITUDE-BASED SPARSIFICATION SCHEME

First, consider the combination of a naive symmetric tensor-wise max-scaled quantization scheme and magnitude-based sparsification scheme at the model level. While the tensor-level analysis is not sufficient for model-level analysis for any combination of quantization and sparsification schemes, it is sufficient in this case.

**Theorem 2.** *Let $L : \mathbb{R}^d \to \mathbb{R}$ be the loss function. Assume that the Hessian $\mathbf{H}_L(\mathbf{w}) = c\mathbb{I}_{d \times d}$. Then, under a naive symmetric tensor-wise max-scaled quantization scheme and tensor-wise magnitude-based sparsification scheme,*

$$L \left( \hat{\mathbf{w}}_{\mathbf{S \to Q}} \right) \leq L \left( \hat{\mathbf{w}}_{\mathbf{Q \to S}} \right) + \mathcal{O}(\| \Delta \mathbf{w_{Q \to S}} \|^3).$$

In this special case, Theorem 1 implies that $\mathbf{S} \to \mathbf{Q}$ obtains approximately better loss than $\mathbf{Q} \to \mathbf{S}$, since the change in loss is (to second-order) proportional to the $L_2$ norm. However, this assumption is very restrictive, and in Section 5.2 we generalize model-level guarantees to the case where $\mathbf{H}_L(\mathbf{w})$ is diagonal. In Section 5.3, we compare the losses when applying $\mathbf{Q} \to \mathbf{S}$ and $\mathbf{S} \to \mathbf{Q}$ in a more general setting.

### 5.2 NAIVE TENSOR-WISE QUANTIZATION SCHEME AND OBD SPARSIFICATION SCHEME

Now, we relax the assumption that $\mathbf{H}_L(\mathbf{w}) = c\mathbb{I}_{d \times d}$. Instead, we assume that $\mathbf{H}_L(\mathbf{w})$ is a diagonal matrix. We consider a naive tensor-wise quantization scheme with a step size of $\delta$ and the OBD sparsification scheme. Optimal Brain Damage (OBD) (LeCun et al., 1989), defines the score function

$$M_i \left( \mathbf{w} \right) \triangleq \frac{1}{2} h_{ii} \mathbf{w}_i^2.$$

We choose the OBD sparsification scheme since it is optimal at the model level assuming that $\mathbf{H}_L(\mathbf{w})$ is a diagonal matrix.

**Theorem 3.** *Let $L : \mathbb{R}^d \to \mathbb{R}$ be the loss function, and assume it is twice differentiable and minimized at the trained weights $\mathbf{w}$. Assume that the Hessian $\mathbf{H}_L(\mathbf{w})$ is diagonal. Then, there exists $\varepsilon(L, \mathbf{w}) > 0$ such that using the naive scaled tensor-wise quantization scheme with a scale bounded by $\delta \leq \varepsilon$ and the OBD sparsification scheme,*

$$L\left(\hat{\mathbf{w}}_{\mathbf{S} \to \mathbf{Q}}\right) \leq L\left(\hat{\mathbf{w}}_{\mathbf{Q} \to \mathbf{S}}\right) + \mathcal{O}(\|\Delta \mathbf{w}_{\mathbf{Q} \to \mathbf{S}}\|^3).$$

This result shows for a fine enough quantization scheme we can show that $\mathbf{S} \to \mathbf{Q}$ is preferred to $\mathbf{Q} \to \mathbf{S}$. To the best of our knowledge, this is the first result that considers the model-level change in loss. If $\mathbf{S} \to \mathbf{Q}$ is a desired ordering, we have shown that this combination of schemes works well. We leave the full proof to Appendix E.

### 5.3 Considering $\mathbf{Q} \to \mathbf{S}$

Above, we have shown quantization and sparsification scheme pairings that prefer $\mathbf{S} \to \mathbf{Q}$. Now, we show that the $\mathbf{Q} \to \mathbf{S}$ ordering need not always be bad, challenging the superiority of $\mathbf{S} \to \mathbf{Q}$ in the general setting. Practically, $\mathbf{Q} \to \mathbf{S}$ may be desired, since models may be too large to be distributed in full precision. In such cases, quantized models can be distributed and additional sparsity may be desired. First, we will prove a theorem to show that $\mathbf{Q} \to \mathbf{S}$ need not always be worse than $\mathbf{S} \to \mathbf{Q}$ under some mild assumptions. Then, in Section 6, we present one such counterexample that supports this argument. With these results, we show it is possible to choose an optimal pairing given a desired ordering.

**Theorem 4.** *Let $\varepsilon_{\mathbf{S}} = (\Delta \mathbf{w}_{\mathbf{Q} \to \mathbf{S}} - \Delta \mathbf{w}_{\mathbf{Q}}) - \Delta \mathbf{w}_{\mathbf{S}}$. Under the assumption that $\Delta \mathbf{w}_{\mathbf{Q}|\mathbf{S}} = \Delta \mathbf{w}_{\mathbf{Q}}$, then*

$$L\left(\hat{\mathbf{w}}_{\mathbf{Q} \to \mathbf{S}}\right) - L\left(\hat{\mathbf{w}}_{\mathbf{S} \to \mathbf{Q}}\right)$$

$$= \varepsilon_{\mathbf{S}}^{\mathsf{T}} \mathbf{H}_L(\mathbf{w})(\Delta \mathbf{w}_{\mathbf{Q}} + \Delta \mathbf{w}_{\mathbf{S}}) + \frac{1}{2}\varepsilon_{\mathbf{S}}^{\mathsf{T}} \mathbf{H}_L(\mathbf{w}) \varepsilon_{\mathbf{S}} + \mathcal{O}\left(\|\Delta \mathbf{w}_{\mathbf{Q} \to \mathbf{S}}\|^3\right) + \mathcal{O}\left(\|\Delta \mathbf{w}_{\mathbf{S} \to \mathbf{Q}}\|^3\right).$$

While strong, the assumption that $\Delta \mathbf{w}_{\mathbf{Q}|\mathbf{S}} = \Delta \mathbf{w}_{\mathbf{Q}}$ is reasonable given current quantization and sparsification schemes. In fact, we will later provide empirical data that gives credence to this. Intuitively, sparsification schemes will try to remove unimportant weights, whereas quantization schemes will attempt to modify the important weights. Thus, the error in quantization before and after sparsification is applied should be similar. In Appendix C, we show that for naive symmetric tensor-wise max-scaled quantization schemes, $\overline{\mathbf{Q}}\left(\overline{\mathbf{S}}(\mathbf{w})\right)_i = \overline{\mathbf{Q}}(\mathbf{w})_i$ assuming that $\mathbf{w}_i$ is not pruned. While our focus in this section is to extend beyond naive symmetric tensor-wise max-scaled quantization schemes, this gives us confidence that the assumptions are reasonable.

We have that $\frac{1}{2}\varepsilon_{\mathbf{S}}^{\mathsf{T}} \mathbf{H}_L(\mathbf{w}) \varepsilon_{\mathbf{S}} \geq 0$, since $\mathbf{H}_L(\mathbf{w})$ is a symmetric matrix, and $\mathbf{w}$ is assumed to be a local minimum, so $\mathbf{H}_L(\mathbf{w})$ is positive semidefinite. However, the sign of $\varepsilon_{\mathbf{S}}^{\mathsf{T}} \mathbf{H}_L(\mathbf{w})(\Delta \mathbf{w}_{\mathbf{Q}} + \Delta \mathbf{w}_{\mathbf{S}})$ is unknown. This leads us to the informal corollary of this result that the difference in losses can be any sign. This implies that there could be instances where approximately $L\left(\hat{\mathbf{w}}_{\mathbf{Q} \to \mathbf{S}}\right) \leq L\left(\hat{\mathbf{w}}_{\mathbf{S} \to \mathbf{Q}}\right)$.

### 5.4 Takeaways

To derive the result in Theorem 4, we apply a second-order Taylor expansion of the loss function $L$ around the point $\mathbf{w}$. The full derivation is deferred to Appendix F, but we highlight the key insights here.

At first glance, this result may appear to contradict a widely held belief in the literature that applying sparsification before quantization ($\mathbf{S} \to \mathbf{Q}$) is universally better than the reverse ordering ($\mathbf{Q} \to \mathbf{S}$). This apparent discrepancy can be reconciled by examining the behavior of typical sparsification algorithms for $\mathbf{Q} \to \mathbf{S}$. When sparsification is applied after quantization, the sparsification algorithm is typically chosen based on its performance on trained weights $\mathbf{w}$. However, the quantized weights $\mathbf{w} + \Delta \mathbf{w}_{\mathbf{Q}}$ have already been perturbed and thus may now lie far from a local minimum. As a result, the standard assumption that the first-order term of the Taylor approximation is negligible no longer holds. In particular, most algorithms implicitly optimize only the second-order term

$$\frac{1}{2}\Delta \mathbf{w}_{\mathbf{S}|\mathbf{Q}}^{\mathsf{T}} \mathbf{H}_L(\mathbf{w} + \Delta \mathbf{w}_{\mathbf{Q}})\Delta \mathbf{w}_{\mathbf{S}|\mathbf{Q}},$$

while ignoring the contribution of the first-order term $\Delta \mathbf{w}_{\mathbf{S}|\mathbf{Q}}^{\mathsf{T}} \nabla L (\mathbf{w} + \Delta \mathbf{w}_{\mathbf{Q}})$ term. Although this omission may be justified when the model is near optimal, it becomes problematic under low-bit quantization, where the quantized model is far from the original optimum. This mismatch introduces a key limitation in sequential compression pipelines. Currently, the quantization and sparsification algorithms are selected independently to be optimal in isolation. They are chosen without consideration for how one transformation might affect the downstream performance of the other. For example, we choose some optimal quantization scheme like AWQ (Lin et al., 2024) or GPTQ (Frantar et al., 2023a), then apply some optimal sparsity scheme like magnitude-based pruning, OBD (LeCun et al., 1989), or OBS (Hassibi & Stork, 1992). However, these are only optimal with respect to the original optimal weights $\mathbf{w}$. Once they are perturbed, the second scheme may no longer be optimal.

This illuminates the following key insight. To ensure favorable performance when combining quantization and sparsification, one must consider their joint effect on model weights. Rather than treating them as independent steps, it may be beneficial to **co-design** the algorithms for quantization and sparsification. This motivates **Quantization-Aware Sparsification**, which we introduce in Section 6 as a counterexample to show that $\mathbf{Q} \rightarrow \mathbf{S}$ may perform better than $\mathbf{S} \rightarrow \mathbf{Q}$ in some cases.

## 6 Quantization-Aware Sparsification (QAS)

The mathematical analysis presented above is general in scope. Specifically, we do not specify how the perturbations $\Delta \mathbf{w}_{\mathbf{Q}}$, $\Delta \mathbf{w}_{\mathbf{S}}$, $\Delta \mathbf{w}_{\mathbf{S}|\mathbf{Q}}$, and $\Delta \mathbf{w}_{\mathbf{Q}|\mathbf{S}}$ are selected. Instead, we solely rely only on the assumption that $\Delta \mathbf{w}_{\mathbf{Q}|\mathbf{S}} = \Delta \mathbf{w}_{\mathbf{Q}}$. This generality allows for a broad theoretical understanding, but it does not give a specific counterexample of a setting where $\mathbf{Q} \rightarrow \mathbf{S}$ may perform better.

To this end, we introduce **Quantization-Aware Sparsification (QAS)**, a novel sparsification scheme explicitly designed with quantization in mind. By tailoring the sparsity pattern to the quantized weights, we can recover a substantial portion of the performance degradation typically incurred under the $\mathbf{Q} \rightarrow \mathbf{S}$ ordering.

For our empirical analysis, we will use a pre-trained OPT-125M (Zhang et al., 2022) as a toy example with the goal of providing a counterexample to the prevailing belief in the literature. OPT-125M is a lightweight model more suitable for deployment on edge devices where resources are constrained. The transformer backbone of OPT-125M makes it representative of other language models, containing key architectural elements such as multi-head attention (Vaswani et al., 2023), layer normalization (Ba et al., 2016), and residual connections (He et al., 2015). Furthermore, it is a common benchmark used for quantization methods (Frantar et al., 2023a; Lin et al., 2024). We test the model on the WikiText (Merity et al., 2016) dataset.

### 6.1 Activation-Aware Quantization and Tensor-Wise Magnitude-Based Sparsification

Now, we will empirically show that choosing quantization and sparsification schemes independently may make it seem as if $\mathbf{S} \rightarrow \mathbf{Q}$ is preferred over $\mathbf{Q} \rightarrow \mathbf{S}$. To that end, we use AWQ (Lin et al., 2024) as our quantization method and tensor-wise magnitude-based pruning as our sparsity method.

We see that under these choices, $\mathbf{S} \rightarrow \mathbf{Q}$ outperforms $\mathbf{Q} \rightarrow \mathbf{S}$ rather significantly for the results without the asterisk. This validates the empirical results seen in the existing literature. For the results with the asterisk, the sparsification threshold following quantization was already $0$. Thus, $\Delta \mathbf{w}_{\mathbf{S}|\mathbf{Q}} = \mathbf{0}$, and this was equivalent to just performing quantization. Therefore, we see that in these cases, $\mathbf{Q} \rightarrow \mathbf{S}$ outperforms $\mathbf{S} \rightarrow \mathbf{Q}$. However, these cases are useful, since it shows us that the assumption $\Delta \mathbf{w}_{\mathbf{Q}|\mathbf{S}} = \Delta \mathbf{w}_{\mathbf{Q}}$ used in Theorem 4 is reasonable, since the performance is comparable between $\overline{\mathbf{Q}}$ and $\overline{\mathbf{Q}} \circ \overline{\mathbf{S}}$.

### 6.2 QAS Formulation

Next, we propose a new magnitude-based sparsity scheme that is "quantization-aware". In particular, the method is best applied when weights have already been quantized.

Table 1: AWQ and Magnitude-Based Results

| PPL↓ | | | | | OPT |
|------|---|---|---|---|-----|
| **Precision** | **Quantization Method** | **Sparsity** | **Sparsity Method** | **Order** | **125M** |
| **FP16** | N/A | 0% | N/A | N/A | 27.656 |
| **INT8** | AWQ | 10% | Magnitude | $\mathbf{S} \to \mathbf{Q}$ | 28.557 |
| **INT8** | AWQ | 10% | Magnitude | $\mathbf{Q} \to \mathbf{S}$ | 99.897 |
| **INT4** | AWQ | 10% | Magnitude | $\mathbf{S} \to \mathbf{Q}$ | 29.913 |
| **INT4** | AWQ | 10% | Magnitude | $\mathbf{Q} \to \mathbf{S}$ | 29.113* |
| **INT8** | AWQ | 25% | Magnitude | $\mathbf{S} \to \mathbf{Q}$ | 35.619 |
| **INT8** | AWQ | 25% | Magnitude | $\mathbf{Q} \to \mathbf{S}$ | 8048.963 |
| **INT4** | AWQ | 25% | Magnitude | $\mathbf{S} \to \mathbf{Q}$ | 37.539 |
| **INT4** | AWQ | 25% | Magnitude | $\mathbf{Q} \to \mathbf{S}$ | 6645.218 |

$^{*} \Delta \mathbf{w}_{\mathbf{S}|\mathbf{Q}} = \mathbf{0}$

### 6.2.1 MOTIVATION

As discussed in the intuition for Theorem 1, the primary cause of suboptimal performance of $\mathbf{Q} \to \mathbf{S}$ is the presence of collisions, which causes distinct weights to become indistinguishable after quantization. To address this issue, we propose a sparsification strategy that aims to recover lost information by leveraging the original, unquantized weights $\mathbf{w}$, even after quantization has been applied. Specifically, we maintain a set of shadow weights of $\mathbf{w}$. During sparsification, we compute the pruning mask based on these shadow weights rather than the quantized ones, but we still apply the mask to the quantized weights.

### 6.2.2 USING THE MAGNITUDE-BASED SPARSITY MASK FROM $\mathbf{w}$

Previously, we defined $M_i(\mathbf{w}) = |\mathbf{w}_i|$ for a magnitude-based sparsification scheme. Now, assuming a quantization scheme has been applied to original weights $\mathbf{w}$ to get quantized weights $\hat{\mathbf{w}}$, we redefine the score function as

$$M_i(\hat{\mathbf{w}}) = |\mathbf{w}_i|.$$

Thus, the score function retrieves the original trained weights before calculating the magnitude. Instead of using the perturbed weights $\hat{\mathbf{w}} = \mathbf{w} + \Delta \mathbf{w}_{\mathbf{Q}}$ to find the mask $\mathbf{m}$, we use the original $\mathbf{w}$ instead.

In other words, we perform the following procedure. We first take pre-trained weights $\mathbf{w}$. Then, we quantize it with some quantization algorithm of choice to get $\hat{\mathbf{w}}$. Finally, we apply $p\%$ magnitude-based sparsity on $\hat{\mathbf{w}}$ using the sparsification scheme outlined above. We show the results in the table below.

Table 2: AWQ and QAS Results

| PPL↓ | | | | | OPT |
|------|---|---|---|---|-----|
| **Precision** | **Quantization Method** | **Sparsity** | **Sparsity Method** | **Order** | **125M** |
| **FP16** | N/A | 0% | N/A | N/A | 27.656 |
| **INT8** | AWQ | 10% | QAS | $\mathbf{Q} \to \mathbf{S}$ | 28.567 |
| **INT4** | AWQ | 10% | QAS | $\mathbf{Q} \to \mathbf{S}$ | 29.864 |
| **INT8** | AWQ | 25% | QAS | $\mathbf{Q} \to \mathbf{S}$ | 35.651 |
| **INT4** | AWQ | 25% | QAS | $\mathbf{Q} \to \mathbf{S}$ | 37.613 |

The results suffice in providing the desired counterexample. Comparing Tables 1 and 2, we see that $\mathbf{Q} \rightarrow \mathbf{S}$ has the ability to outperform $\mathbf{S} \rightarrow \mathbf{Q}$ given an appropriate choice of sparsity algorithm.

### 6.3 REMOVING THE DEPENDENCY ON $\mathbf{w}$

As noted above, QAS was introduced primarily as a counterexample to existing literature. As such, the proposed sparsification scheme is admittedly naive. It requires retaining a full-precision copy of the weights $\mathbf{w}$, or equivalently $\Delta \mathbf{w_Q}$, even after quantization. As discussed in our motivation, this contradicts the goal of fully eliminating the dependency on high-precision representations. In this section, we show that a more reasonable sparsification scheme is attainable from QAS.

First, observe that retaining full-precision values of $\mathbf{w}$ is not strictly necessary for QAS. Only their relative orderings are required to determine which weights to prune. Therefore, instead of storing the actual floating-point weights of the original weights $\mathbf{w}$, we can store a ranking or ordering of the weight indices based on their original magnitudes. This representation is significantly more compact and avoids the overhead of retaining the full-precision tensor.

Alternatively, we can remove the dependence on $\mathbf{w}$ altogether through minimal retraining. For some quantized weight $(\hat{\mathbf{w}}_{\mathbf{Q}})_i$, the main goal is to decide whether the original weight $\mathbf{w}_i = (\hat{\mathbf{w}}_{\mathbf{Q}})_i - (\Delta \mathbf{w_Q})_i$ was pruned. A naive way to find this is to re-train the quantized model to the original optimal $\mathbf{w}$. However, as stated previously, this would be time-consuming and undesirable.

Instead, we make the following key insight. We are only concerned with disambiguating collisions, where $|\mathbf{w}_i| < |\mathbf{w}_j|$ but $\left|(\hat{\mathbf{w}}_{\mathbf{Q}})_i\right| = \left|(\hat{\mathbf{w}}_{\mathbf{Q}})_j\right|$. Regardless of quantization scheme, we expect that the quantized value will likely share the same sign as the original value. Thus, it is sufficient to consider the case where $\mathbf{w}_i$ and $\mathbf{w}_j$ share the same sign, so $(\hat{\mathbf{w}}_{\mathbf{Q}})_i = (\hat{\mathbf{w}}_{\mathbf{Q}})_j$, like in Appendix C. Since we just need to separate these collided quantized weights, we need not train the model all the way to convergence. Instead, we can perform gradient updates $\hat{\mathbf{w}}_{t+1} = \hat{\mathbf{w}}_t - \eta \nabla_{\mathbf{w}} L$ until the indistinguishable weights become distinct again. The direction of the update will generally shift $\hat{\mathbf{w}}_{\mathbf{Q}}$ towards $\mathbf{w}$ and "undoes" the rounding performed by quantization. Note we can replace gradient descent with any iterative method, such as stochastic gradient descent; the main insight is that since we only care about comparing $|\mathbf{w}_i|$ and $|\mathbf{w}_j|$, thus we can stop performing gradient updates substantially earlier.

## 7 CONCLUSION

In this work, we provide further evidence that quantization and sparsification are not orthogonal operations. At the tensor level, we prove that applying sparsification before quantization consistently yields lower $L_p$ reconstruction error for all $p \geq 1$ under common schemes. However, we also show that such tensor-level metrics are insufficient to predict overall model performance, motivating the need for model-level analysis.

To this end, we present the first theoretical results demonstrating that, in standard settings where quantization and sparsification algorithms are chosen independently, the ordering $\mathbf{S} \rightarrow \mathbf{Q}$ achieves approximately lower model loss than $\mathbf{Q} \rightarrow \mathbf{S}$. Notably, we show that this preference does not hold universally though. $\mathbf{Q} \rightarrow \mathbf{S}$ can perform comparably, or even better, than $\mathbf{S} \rightarrow \mathbf{Q}$, depending on the compression strategy. To illustrate this, we introduce Quantization-Aware Sparsification (QAS), a novel sparsification scheme that is designed to be applied to quantized models. Then, using QAS, we present a counterexample to show that $\mathbf{Q} \rightarrow \mathbf{S}$ could perform better.

Our findings underscore the importance of co-designing quantization and sparsification algorithms, rather than treating them as isolated steps. We hope this work encourages further research into unified compression frameworks that account for the interactions between different compression techniques.

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

# A  NAIVE SYMMETRIC MAX-SCALED BLOCK-WISE QUANTIZATION

In this section, we rigorously define the idea of *naive symmetric max-scaled block-wise quantization* as defined in Gholami et al. (2021).

## A.1  BLOCK-WISE

Suppose we divide the weights into blocks (groups) $\{\mathbf{w}_{(1)}, \ldots, \mathbf{w}_{(B)}\}$, defined by a partition $\{P_1, \ldots, P_B\}$ of $[d]$. Typically, blocks are collections of rows or columns. That is, in order for a quantization scheme to be block-wise, we require that $\mathcal{Q}_i = \mathcal{Q}_j$ and $D_i = D_j$ for all $i, j \in P_k$, where $k \in [B]$. Intuitively, this requires that the dequantization functions be the same for all weights in a block. This is desirable for hardware purposes, since nearby elements will share the same dequantization operations. If $B = 1$, we call this a *tensor-wise* quantization scheme.

## A.2  MAX-SCALED

We define a quantization scheme to be *max-scaled* with respect to blocks $\{\mathbf{w}_{(1)}, \ldots, \mathbf{w}_{(B)}\}$ iff the following conditions hold.

1. $\Theta = \mathbb{R}^B$

2. For all $k \in [B]$ and $i \in P_k$, $\Phi$ is defined by $\left(\Phi\left(\mathbf{w}\right)\right)_i = \max_{j \in P_k} |\mathbf{w}_j|$.

3. For all $k \in [B]$ and $i \in P_k$, we require that $D_i$ is defined as $(q, \theta) \mapsto q\theta_k$.

By the last condition, we can see that max-scaled quantization schemes are also block-wise. We restrict the parameters to be the maximum elements of each block. Then, we require that the dequantization simply scale this maximum. In other words, the quantization grid (range of $D_i$) is parameterized only by the maximum magnitude for an element in a block. The grid scales with the maximum magnitude element. Removing the second condition will give us a *scaled* quantization scheme.

## A.3  NAIVE

Finally, we define a naive quantization scheme. A quantization scheme is *naive* iff

$$Q_i\left(\mathbf{w}, \theta\right) = \arg\min_{a \in \mathcal{Q}_i} |\mathbf{w}_i - D_i\left(a, \theta\right)|$$

for all $\mathbf{w} \in \mathbb{R}^d$ and $\theta \in \Theta$. In particular, this quantizes $\mathbf{w}_i$ to the nearest point in the quantization grid. This is also known as a rounding-to-nearest scheme. Note that under this naive assumption, $Q_i$ is only a function of $\mathbf{w}_i$. In particular, there exists a function $\tilde{Q}_i : \mathbb{R} \times \Theta \to \mathcal{Q}_i$ such that

$$\tilde{Q}_i\left(\mathbf{w}_i, \theta\right) = Q_i\left(\mathbf{w}, \theta\right)$$

for all $\mathbf{w} \in \mathbb{R}^d$ and $\theta \in \Theta$.

## A.4  SYMMETRIC

We call a quantization scheme *symmetric* iff the following three conditions hold.

1. For all $i \in [d]$, $0 \in \mathcal{Q}_i$.

2. For all $i \in [d]$, $\mathcal{Q}_i = -\mathcal{Q}_i$.

3. For all $i \in [d]$, $D_i\left(q, \theta\right) = -D_i\left(-q, \theta\right)$.

Intuitively, the quantization grid is symmetric iff the set of levels is symmetric about zero, and the corresponding dequantization function preserves this symmetry.

## A.5 ORDER-PRESERVING PROPERTY

Now, we illuminate a key property of naive scaled block-wise quantization, which we will call the *order-preserving property*.

**Theorem 5.** *Let $i, j \in P_k$ for some block $k$. Furthermore, suppose $\overline{\mathbf{Q}}$ is a naive scaled block-wise quantization scheme. Then, if $\mathbf{w}_i \leq \mathbf{w}_j$, we then claim that $\overline{\mathbf{Q}}(\mathbf{w})_i \leq \overline{\mathbf{Q}}(\mathbf{w})_j$.*

*Proof.* We want to show that

$$\overline{\mathbf{Q}}(\mathbf{w})_i \leq \overline{\mathbf{Q}}(\mathbf{w})_j.$$

By definition, we have the following.

$$\overline{\mathbf{Q}}(\mathbf{w})_i \leq \overline{\mathbf{Q}}(\mathbf{w})_j$$
$$\mathbf{D}(\mathbf{Q}(\mathbf{w}, \Phi(\mathbf{w})), \Phi(\mathbf{w}))_i \leq \mathbf{D}(\mathbf{Q}(\mathbf{w}, \Phi(\mathbf{w})), \Phi(\mathbf{w}))_j$$
$$D_i(\mathbf{Q}(\mathbf{w}, \Phi(\mathbf{w}))_i, \Phi(\mathbf{w})) \leq D_j\left(\mathbf{Q}(\mathbf{w}, \Phi(\mathbf{w}))_j, \Phi(\mathbf{w})\right)$$
$$D_i(Q_i(\mathbf{w}, \Phi(\mathbf{w})), \Phi(\mathbf{w})) \leq D_j(Q_j(\mathbf{w}, \Phi(\mathbf{w})), \Phi(\mathbf{w}))$$

First, note that $D_i = D_j$ since $i$ and $j$ are in the same partition by the block-wise assumption. Thus, it suffices to show that $D_i(Q_i(\mathbf{w}, \Phi(\mathbf{w})), \Phi(\mathbf{w})) \leq D_j(Q_j(\mathbf{w}, \Phi(\mathbf{w})), \Phi(\mathbf{w}))$. Next, note the following by the scaled assumption.

$$D_i(Q_i(\mathbf{w}, \Phi(\mathbf{w})), \Phi(\mathbf{w})) = Q_i(\mathbf{w}, \Phi(\mathbf{w})) \Phi(\mathbf{w})_k$$

Similarly, note that

$$D_j(Q_j(\mathbf{w}, \Phi(\mathbf{w})), \Phi(\mathbf{w})) = Q_j(\mathbf{w}, \Phi(\mathbf{w})) \Phi(\mathbf{w})_k.$$

Since $\max_{\ell \in P_k} |\mathbf{w}_\ell| \geq 0$, it suffices to show that

$$Q_i(\mathbf{w}, \Phi(\mathbf{w})) \leq Q_j(\mathbf{w}, \Phi(\mathbf{w})).$$

Finally, note that by the naive property, we have that

$$Q_i(\mathbf{w}, \Phi(\mathbf{w})) = \arg\min_{a \in \mathcal{Q}_i} |\mathbf{w}_i - D_i(a, \Phi(\mathbf{w}))|$$
$$Q_j(\mathbf{w}, \Phi(\mathbf{w})) = \arg\min_{a \in \mathcal{Q}_i} |\mathbf{w}_j - D_i(a, \Phi(\mathbf{w}))|.$$

Suppose towards a contradiction that $Q_j(\mathbf{w}, \Phi(\mathbf{w})) < Q_i(\mathbf{w}, \Phi(\mathbf{w}))$. Then, we have that

$$\arg\min_{a \in \mathcal{Q}_i} |\mathbf{w}_j - D_i(a, \Phi(\mathbf{w}))| < \arg\min_{a \in \mathcal{Q}_i} |\mathbf{w}_i - D_i(a, \Phi(\mathbf{w}))|.$$

Let $a_i = \arg\min_{a \in \mathcal{Q}_i} |\mathbf{w}_i - D_i(a, \Phi(\mathbf{w}))|$ and $a_j = \arg\min_{a \in \mathcal{Q}_i} |\mathbf{w}_j - D_i(a, \Phi(\mathbf{w}))|$. Furthermore, we have that

$$D_i(a, \Phi(\mathbf{w})) = a\Phi(\mathbf{w})_k.$$

Let $s_k = a\Phi(\mathbf{w})_k$. We claim that

$$|\mathbf{w}_j - a_j s_k| \geq |\mathbf{w}_j - a_i s_k|.$$

We have that $a_j < a_i$ (by assumption), $\mathbf{w}_i < \mathbf{w}_j$, and $|\mathbf{w}_i - a_i s_k| \leq |\mathbf{w}_i - a_j s_k|$. We have the following.

$$|\mathbf{w}_i - a_j s_k| \geq |\mathbf{w}_i - a_i s_k|$$
$$(\mathbf{w}_i - a_j s_k)^2 \geq (\mathbf{w}_i - a_i s_k)^2$$
$$\mathbf{w}_i^2 - 2\mathbf{w}_i a_j s_k + a_j^2 s_k^2 \geq \mathbf{w}_i^2 - 2\mathbf{w}_i a_i s_k + a_i^2 s_k^2$$
$$-2\mathbf{w}_i a_j s_k + a_j^2 s_k^2 \geq -2\mathbf{w}_i a_i s_k + a_i^2 s_k^2$$
$$2\mathbf{w}_i a_i s_k - 2\mathbf{w}_i a_j s_k \geq a_i^2 s_k^2 - a_j^2 s_k^2$$
$$2\mathbf{w}_i (a_i s_k - a_j s_k) \geq (a_i s_k + a_j s_k)(a_i s_k - a_j s_k)$$
$$2\mathbf{w}_i \geq (a_j s_k + a_i s_k) \qquad (a_j < a_i)$$
$$\mathbf{w}_i \geq \frac{a_i s_k + a_j s_k}{2}$$
$$\mathbf{w}_j \geq \frac{a_i s_k + a_j s_k}{2} \qquad (\mathbf{w}_i < \mathbf{w}_j)$$
$$2\mathbf{w}_j \geq a_i s_k + a_j s_k$$
$$2\mathbf{w}_j (a_i s_k - a_j s_k) \geq (a_i s_k + a_j s_k)(a_i s_k - a_j s_k)$$
$$(\mathbf{w}_j - a_j s_k)^2 \geq (\mathbf{w}_j - a_i s_k)^2$$
$$|\mathbf{w}_j - a_j s_k| \geq |\mathbf{w}_j - a_i s_k|$$

Thus, we have a contradiction, since we assumed $a_j = \underset{a \in \mathcal{Q}_i}{\arg\min} |\mathbf{w}_j - D_i(a, \Phi(\mathbf{w}))|$. $\qquad \square$

We also have the following corollary that is a natural extension of Theorem 5 adding the stronger symmetric property.

**Corollary 6.** *Let $i, j \in P_k$ for some block $k$. Furthermore, suppose $\overline{\mathbf{Q}}$ is a naive symmetric scaled block-wise quantization scheme. Then, if $|\mathbf{w}_i| \leq |\mathbf{w}_j|$, then*

$$\left|\overline{\mathbf{Q}}(\mathbf{w})_i\right| \leq \left|\overline{\mathbf{Q}}(\mathbf{w})_j\right|.$$

*Proof.* Before, we prove the corollary, we prove the following lemmas.

**Lemma 7.** *Suppose $\overline{\mathbf{Q}}$ is a naive symmetric scaled block-wise quantization scheme. Then, if $\mathbf{w}_i = 0$, then $\overline{\mathbf{Q}}(\mathbf{w})_i = 0$.*

*Proof.* Since $\overline{\mathbf{Q}}$ is symmetric, we have that $0 \in \mathcal{Q}_i$ for all $i \in [d]$. Furthermore, since $\overline{\mathbf{Q}}$ is scaled, we have that $D_i(0, \theta) = 0$ for all $i \in [d]$ and $\theta \in \Theta$. Therefore, we have that $Q_i(\mathbf{w}, \theta) = 0$ for all $i \in [d]$ and $\theta \in \Theta$ by the naive assumption. Thus, we have that $\mathbf{Q}(\mathbf{w}, \Phi(\mathbf{w}))_i = 0$ and $\overline{\mathbf{Q}}(\mathbf{w})_i = \mathbf{D}(\mathbf{Q}(\mathbf{w}, \Phi(\mathbf{w})), \Phi(\mathbf{w}))_i = 0.$ $\qquad \square$

**Lemma 8.** *Suppose $\overline{\mathbf{Q}}$ is a naive symmetric scaled block-wise quantization scheme. Construct $\tilde{\mathbf{w}} \in \mathbb{R}^d$. Let $\tilde{\mathbf{w}}_i = -\mathbf{w}_i$ and $\tilde{\mathbf{w}}_j = \mathbf{w}_j$ for all $i \neq j$. Then,*

$$\left|\overline{\mathbf{Q}}(\mathbf{w})_k\right| = \left|\overline{\mathbf{Q}}(\tilde{\mathbf{w}})_k\right|$$

*for all $k \in [d]$.*

*Proof.* We have that $\Phi(\tilde{\mathbf{w}}) = \Phi(\mathbf{w})$. First, consider $k \neq i$. Then, we have that $\overline{\mathbf{Q}}(\mathbf{w})_k = \overline{\mathbf{Q}}(\tilde{\mathbf{w}})_k$, so $\left|\overline{\mathbf{Q}}(\mathbf{w})_k\right| = \left|\overline{\mathbf{Q}}(\tilde{\mathbf{w}})_k\right|$, since $Q_i$ and $D_i$ are only functions of $\mathbf{w}_i$ and $\Phi(\mathbf{w})$ by our naive and scaled assumption.

Now, consider $k = i$. We have the following.

$$\left|\overline{\mathbf{Q}}(\tilde{\mathbf{w}})_i\right| = |\mathbf{D}(\mathbf{Q}(\tilde{\mathbf{w}}, \Phi(\tilde{\mathbf{w}})), \Phi(\tilde{\mathbf{w}}))_i|$$
$$= |\mathbf{D}(\mathbf{Q}(\tilde{\mathbf{w}}, \Phi(\mathbf{w})), \Phi(\mathbf{w}))_i| \qquad (\Phi(\mathbf{w}) = \Phi(\tilde{\mathbf{w}}))$$
$$= |D_i(\mathbf{Q}(\tilde{\mathbf{w}}, \Phi(\mathbf{w}))_i, \Phi(\mathbf{w}))|$$
$$= |D_i(Q_i(\tilde{\mathbf{w}}, \Phi(\mathbf{w})), \Phi(\mathbf{w}))|$$

By the naive assumption, we have that

$$Q_i \left( \mathbf{w}, \theta \right) = \underset{a \in \mathcal{Q}_i}{\arg \min} \left| \mathbf{w}_i - D_i \left( a, \theta \right) \right|$$

and

$$Q_i \left( \tilde{\mathbf{w}}, \theta \right) = \underset{a \in \mathcal{Q}_i}{\arg \min} \left| - \mathbf{w}_i - D_i \left( a, \theta \right) \right|.$$

Then, we have the following.

$$
\begin{aligned}
Q_i \left( \tilde{\mathbf{w}}, \theta \right) &= \underset{a \in \mathcal{Q}_i}{\arg \min} \left| - \mathbf{w}_i - D_i \left( a, \theta \right) \right| \\
&= \underset{-a \in \mathcal{Q}_i}{\arg \min} \left| - \mathbf{w}_i - D_i \left( -a, \theta \right) \right| & (\mathcal{Q}_i = -\mathcal{Q}_i) \\
&= \underset{-a \in \mathcal{Q}_i}{\arg \min} \left| - \mathbf{w}_i + D_i \left( a, \theta \right) \right| & (D_i \left( q, \theta \right) = -D_i \left( -q, \theta \right)) \\
&= \underset{-a \in \mathcal{Q}_i}{\arg \min} \left| \mathbf{w}_i - D_i \left( a, \theta \right) \right| \\
&= -Q_i \left( \mathbf{w}, \theta \right)
\end{aligned}
$$

Thus, we have the following.

$$
\begin{aligned}
\left| \overline{\mathbf{Q}} \left( \tilde{\mathbf{w}} \right)_i \right| &= \left| D_i \left( -Q_i \left( \mathbf{w}, \Phi \left( \mathbf{w} \right) \right), \Phi \left( \mathbf{w} \right) \right) \right| \\
&= \left| -D_i \left( Q_i \left( \mathbf{w}, \Phi \left( \mathbf{w} \right) \right), \Phi \left( \mathbf{w} \right) \right) \right| & (D_i \left( q, \theta \right) = -D_i \left( -q, \theta \right)) \\
&= \left| D_i \left( Q_i \left( \mathbf{w}, \Phi \left( \mathbf{w} \right) \right), \Phi \left( \mathbf{w} \right) \right) \right| \\
&= \left| \overline{\mathbf{Q}} \left( \mathbf{w} \right)_i \right|
\end{aligned}
$$

$\square$

There are four cases to consider. First, consider the case when $0 \leq \mathbf{w}_i < \mathbf{w}_j$. Then, by Theorem 5 and Lemma 7, we have that $0 \leq \overline{\mathbf{Q}} \left( \mathbf{w} \right)_i \leq \overline{\mathbf{Q}} \left( \mathbf{w} \right)_j$. Thus, $\left| \overline{\mathbf{Q}} \left( \mathbf{w} \right)_i \right| \leq \left| \overline{\mathbf{Q}} \left( \mathbf{w} \right)_j \right|$.

Next, consider the case where $\mathbf{w}_j \leq \mathbf{w}_i \leq 0$. Again, by Theorem 5 and Lemma 7, we have that $\overline{\mathbf{Q}} \left( \mathbf{w}_j \right) \leq \overline{\mathbf{Q}} \left( \mathbf{w}_i \right) \leq 0$. Thus, $\left| \overline{\mathbf{Q}} \left( \mathbf{w}_i \right) \right| \leq \left| \overline{\mathbf{Q}} \left( \mathbf{w}_j \right) \right|$.

Next, consider the case where $\mathbf{w}_i \leq 0 \leq \mathbf{w}_j$. Then, construct $\tilde{\mathbf{w}}$ as described in Lemma 8. We have that $0 \leq -\mathbf{w}_i \leq \mathbf{w}_j$, so $0 \leq \tilde{\mathbf{w}}_i \leq \tilde{\mathbf{w}}_j$. Then, by the first case and Lemma 8, we have that

$$\left| \overline{\mathbf{Q}} \left( \mathbf{w} \right)_i \right| = \left| \overline{\mathbf{Q}} \left( \tilde{\mathbf{w}} \right)_i \right| < \left| \overline{\mathbf{Q}} \left( \tilde{\mathbf{w}} \right)_j \right| = \left| \overline{\mathbf{Q}} \left( \mathbf{w} \right)_j \right|.$$

Finally, consider the case where $\mathbf{w}_j \leq 0 \leq \mathbf{w}_i$. Then, construct $\tilde{\mathbf{w}}$ as described in Lemma 8. We have that $\mathbf{w}_j \leq -\mathbf{w}_i \leq 0$, so $\tilde{\mathbf{w}}_j \leq \tilde{\mathbf{w}}_i \leq 0$. Then, by the second case and Lemma 8, we have that

$$\left| \overline{\mathbf{Q}} \left( \mathbf{w} \right)_i \right| = \left| \overline{\mathbf{Q}} \left( \tilde{\mathbf{w}} \right)_i \right| < \left| \overline{\mathbf{Q}} \left( \tilde{\mathbf{w}} \right)_j \right| = \left| \overline{\mathbf{Q}} \left( \mathbf{w} \right)_j \right|.$$

$\square$

### A.6 ALTERNATIVE QUANTIZATION SCHEMES

However, we should note that most modern quantization schemes are not naive symmetric max-scaled block-wise quantization schemes. AdaRound (Nagel et al., 2020) was an early alternative. In particular, AdaRound violates the naive assumption, since it does not employ rounding-to-nearest. Instead, it solves a binary optimization problem to decide whether to round $\mathbf{w}_i$ "up" or "down". In particular,

$$Q_i \left( \mathbf{w}, \theta \right) \neq \underset{a \in \mathcal{Q}_i}{\arg \min} \left| \mathbf{w}_i - D_i \left( a, \theta \right) \right|.$$

This may round in the opposite direction of the naive assumption and violates the order-preserving property. In particular, since we decide whether to round up or round down on each weight, it is possible that $\mathbf{w}_i$ rounds up and $\mathbf{w}_j$ rounds down, even if $\mathbf{w}_i < \mathbf{w}_j$.

Recently, Activation-Aware Quantization (AWQ) (Lin et al., 2024) has emerged as a popular quantization scheme for LLMs. AWQ also violates the naive assumption. We formalize the AWQ quantization scheme presented in Lin et al. (2024) and show that it is not naive. We start with a naive symmetric max-scaled block-wise quantization scheme $\overline{\mathbf{Q}}$. Then, we will construct the AWQ quantization scheme, $\widetilde{\overline{\mathbf{Q}}}$.

Define a new $\widetilde{\Phi}\left(\mathbf{w}\right) = \begin{bmatrix} \Phi\left(\mathbf{w}\right) \\ \mathbf{r} \end{bmatrix}$, where $\mathbf{r} \in \mathbb{R}^{d'}$. $\mathbf{r}$ is defined in Lin et al. (2024). Then, we define

$$\widetilde{Q}_i\left(\mathbf{w}, \theta\right) = Q_i\left(\text{diag}\left(\mathbf{r}\right)\mathbf{w}, \mathbf{s}\right),$$

where $\theta = \begin{bmatrix} \mathbf{s} \\ \mathbf{r} \end{bmatrix}$. Then, we define

$$\widetilde{D}_i\left(q, \theta\right) = D_i\left(q, \theta\right).$$

AWQ's quantization strategy is unique in that there are two different scaling factors, $\mathbf{r}$ and $\mathbf{s}$. A group of weights is still on the same quantization grid, but each of the rows will have different scaling factors applied to it. Therefore, even if rounding-to-nearest is used for the original quantization grid, the additional scaling factor may break the order-preserving property. In particular, let $\mathbf{w}_i < \mathbf{w}_j$. Suppose that these weights are in the same group, but in different rows with scaling factors $r_i$ and $r_j$ respectively. Then, let

$$\left\lfloor \frac{r_i \mathbf{w}_i}{s} \right\rceil = \left\lfloor \frac{r_j \mathbf{w}_j}{s} \right\rceil,$$

where $s$ is the scaling factor for the group. Then, it is possible that

$$\frac{s}{r_i}\left\lfloor \frac{r_i \mathbf{w}_i}{s} \right\rceil > \frac{s}{r_j}\left\lfloor \frac{r_j \mathbf{w}_j}{s} \right\rceil.$$

In particular, let $r_i = 2$, $\mathbf{w}_i = 0.4$, $r_j = 1$, $\mathbf{w}_j = 0.45$, and $s = 1$. Then, we have

$$\frac{s}{r_i}\left\lfloor \frac{r_i \mathbf{w}_i}{s} \right\rceil = \frac{1}{2}$$

and

$$\frac{s}{r_j}\left\lfloor \frac{r_j \mathbf{w}_j}{s} \right\rceil = 0,$$

assuming $\mathcal{Q}_i = \mathcal{Q}_j = \mathbb{Z}$. Thus, AWQ is not a naive symmetric max-scaled block-wise quantization scheme.

## B  MAGNITUDE-BASED BLOCK-WISE SPARSIFICATION

We now rigorously define *magnitude-based block-wise sparsification* as defined in Hoefler et al. (2021).

### B.1  BLOCK-WISE

We can define a "block-wise" sparsification scheme is as follows. Let $\{M_i\}_{i=1}^d$ consists of a set of score (or saliency) functions $M_i : \mathbb{R}^d \to \mathbb{R}$ for each weight $\mathbf{w}_i$. The intuition is that we can calculate the score for each weight $\mathbf{w}_i$ and prune those that have the smallest score with some blocks. Again, like block-wise quantization, divide the weights into blocks $\{\mathbf{w}_{(1)}, \ldots, \mathbf{w}_{(B)}\}$, defined by a partition $\{P_1, \ldots, P_B\}$ of $[d]$. Like in quantization, if $B = 1$, we call this a *tensor-wise* sparsification scheme.

Now, consider the set $T_k = \{M_i\left(\mathbf{w}\right) : i \in P_k\}$ for all $k \in [B]$. Then, to achieve $p\%$ sparsity for $P_k$, we select some $t_k \geq p\% \cdot |P_k|$. Let $T_k^{(t_k)}$ be the $t_k$th element when $T_k$ is sorted by magnitude. Then, for all $M_i\left(\mathbf{w}\right) \leq T_k^{(t_k)}$, we want to map the weight $\mathbf{w}_i$ to 0. Thus, for $i \in P_k$, we can define $S_i : \mathbb{R}^d \to \{0, 1\}$ such that

$$S_i\left(\mathbf{w}\right) = \begin{cases} 0 & M_i\left(\mathbf{w}\right) \leq T_k^{(t_k)} \\ 1 & \text{otherwise.} \end{cases}.$$

We have that this is a valid $p\%$ sparsification scheme, since

$$\left|\{S_i\left(\mathbf{w}\right)=0 \mid i \in [d]\}\right| = \sum_{k=1}^{B} \left|\left\{M_i\left(\mathbf{w}\right) \leq T_k^{(t_k)} \mid i \in P_k\right\}\right| \geq \sum_{k=1}^{B} p\% \cdot |P_k| \geq p\% \cdot d.$$

## B.2 MAGNITUDE-BASED

As the name suggests, the magnitude-based sparsification scheme uses weight magnitude as a metric of saliency. Using a block-wise sparsification scheme, we can simply define $M_i : \mathbb{R}^d \rightarrow \mathbb{R}$ as $\mathbf{w} \mapsto |\mathbf{w}_i|$. Intuitively, we set a block-wise threshold and prune all the weights with magnitude less than that threshold in that block. It is known that under the assumption that $\mathbf{H}_L(\mathbf{w}) = \mathbb{I}_{d \times d}$ that the magnitude-based sparsification scheme is optimal.

## B.3 ALTERNATIVE SPARSIFICATION SCHEMES

As mentioned above, the magnitude-based sparsification scheme relies on the assumption that the Hessian is the identity matrix. Thus, more sophisticated sparsification schemes have been developed. For example, Optimal Brain Damage (OBD) (LeCun et al., 1989), assumes that the Hessian is diagonal, defining the score function

$$M_i\left(\mathbf{w}\right) \triangleq \frac{1}{2} h_{ii} \mathbf{w}_i^2,$$

where $h_{ii} = \frac{\partial^2 L}{\partial \mathbf{w}_i^2}$ is the $i$th diagonal element of the Hessian matrix. Optimal Brain Surgeon (OBS) (Hassibi & Stork, 1992) and Optimal Brian Compression (OBC) (Frantar et al., 2023b) are two variations of this. Again, these non-examples motivate the need for a more general framework for analyzing these problems.

## C PROOF OF THEOREM 1

*Proof.* We first prove the desired result for finite $p$. We have that

$$\left\|\mathbf{w} - \overline{\mathbf{Q}}\left(\overline{\mathbf{S}}\left(\mathbf{w}\right)\right)\right\|_p = \left(\sum_{i=1}^{d} \left|\mathbf{w}_i - \overline{\mathbf{Q}}\left(\overline{\mathbf{S}}\left(\mathbf{w}\right)\right)_i\right|^p\right)^{\frac{1}{p}}$$

$$\left\|\mathbf{w} - \overline{\mathbf{S}}\left(\overline{\mathbf{Q}}\left(\mathbf{w}\right)\right)\right\|_p = \left(\sum_{i=1}^{d} \left|\mathbf{w}_i - \overline{\mathbf{S}}\left(\overline{\mathbf{Q}}\left(\mathbf{w}\right)\right)_i\right|^p\right)^{\frac{1}{p}}.$$

First, we consider the degenerate case where all weights are pruned. Then, we have that $\overline{\mathbf{S}}\left(\mathbf{w}\right) = \mathbf{0}$. By Lemma 7, we have that $\overline{\mathbf{Q}}\left(\overline{\mathbf{S}}\left(\mathbf{w}\right)\right) = \overline{\mathbf{Q}}\left(\mathbf{0}\right) = \mathbf{0}$. Now, we show that $\overline{\mathbf{S}}\left(\overline{\mathbf{Q}}\left(\mathbf{w}\right)\right) = \mathbf{0}$. For $\overline{\mathbf{S}}$ applied to $\mathbf{w}$, let the threshold be $|\mathbf{w}_t|$ for some $t \in [d]$. Then, we have that

$$\left|\{|\mathbf{w}_i| \leq |\mathbf{w}_t| : i \in [d]\}\right| = \left|\left\{\left|\overline{\mathbf{Q}}\left(\mathbf{w}\right)_i\right| \leq \left|\overline{\mathbf{Q}}\left(\mathbf{w}\right)_t\right| : i \in [d]\right\}\right|$$

by Corollary 6. Thus, we have that $\overline{\mathbf{S}}\left(\overline{\mathbf{Q}}\left(\mathbf{w}\right)\right) = \mathbf{0}$ as desired.

Now, consider the case where there exists an element that is not pruned. First, we have that $\Phi\left(\mathbf{w}\right) = \Phi\left(\overline{\mathbf{S}}\left(\mathbf{w}\right)\right)$, since the element with the largest magnitude is not pruned. Then, for $\overline{\mathbf{Q}}\left(\overline{\mathbf{S}}\left(\mathbf{w}\right)\right)_i$, we have that

$$\overline{\mathbf{Q}}\left(\overline{\mathbf{S}}\left(\mathbf{w}\right)\right)_i = \overline{\mathbf{Q}}\left(\mathbf{w}\right)_i$$

if $\mathbf{w}_i$ is not pruned. This follows from the fact that $Q_i$ and $D_i$ are only functions of $\mathbf{w}_i$ and $\Phi\left(\mathbf{w}\right)$ by our naive and max-scaled assumption. If $\mathbf{w}_i$ is pruned, then we have that $\overline{\mathbf{Q}}\left(\overline{\mathbf{S}}\left(\mathbf{w}\right)\right)_i = 0$, since $0 \in \mathcal{Q}_i$ for all $i \in [d]$. Thus, we have that

$$\left|\mathbf{w}_i - \overline{\mathbf{Q}}\left(\overline{\mathbf{S}}\left(\mathbf{w}\right)\right)_i\right| = \begin{cases} \left|\mathbf{w}_i - \overline{\mathbf{Q}}\left(\mathbf{w}\right)_i\right| & \text{if } \mathbf{w}_i \text{ is not pruned} \\ \left|\mathbf{w}_i - \overline{\mathbf{S}}\left(\mathbf{w}\right)_i\right| & \text{otherwise.} \end{cases}$$

Using this, we will prove the statement

$$\sum_{i \in T} \left| \mathbf{w}_i - \overline{\mathbf{Q}} \left( \overline{\mathbf{S}} \left( \mathbf{w} \right) \right)_i \right|^p \leq \sum_{i \in T} \left| \mathbf{w}_i - \overline{\mathbf{S}} \left( \overline{\mathbf{Q}} \left( \mathbf{w} \right) \right)_i \right|^p$$

inductively on the size of set $T \subseteq [d]$. Let $n = |T|$.

First, we consider the base case where $n = 0$. Then, we have that the statement is trivially true. Now, we show the inductive step. Suppose that the statement is true for some $n = k$. Then, we will show that the statement is true for $n = k + 1$. Consider some set $T$ with cardinality $k + 1$.

First, consider the simpler case. In particular, consider the case where the same elements are pruned before and after quantization. Then, the elements that $\overline{\mathbf{S}}$ prunes remains the same regardless if it is applied to $\mathbf{w}$ or $\overline{\mathbf{Q}} \left( \mathbf{w} \right)$. First, suppose that $\mathbf{w}_i$ is pruned. Then, we have that

$$\left| \mathbf{w}_i - \overline{\mathbf{Q}} \left( \overline{\mathbf{S}} \left( \mathbf{w} \right) \right)_i \right| = \left| \mathbf{w}_i - \overline{\mathbf{S}} \left( \overline{\mathbf{Q}} \left( \mathbf{w} \right) \right)_i \right| = |\mathbf{w}_i| .$$

Suppose $\mathbf{w}_i$ is not pruned. Then,

$$\left| \mathbf{w}_i - \overline{\mathbf{Q}} \left( \overline{\mathbf{S}} \left( \mathbf{w} \right) \right)_i \right| = \left| \mathbf{w}_i - \overline{\mathbf{S}} \left( \overline{\mathbf{Q}} \left( \mathbf{w} \right) \right)_i \right| = \left| \mathbf{w}_i - \overline{\mathbf{Q}} \left( \mathbf{w} \right)_i \right| .$$

This is true for all $i \in [d]$. Therefore, we have that

$$\left\| \mathbf{w} - \overline{\mathbf{Q}} \left( \overline{\mathbf{S}} \left( \mathbf{w} \right) \right) \right\|_p = \left\| \mathbf{w} - \overline{\mathbf{S}} \left( \overline{\mathbf{Q}} \left( \mathbf{w} \right) \right) \right\|_p .$$

Now, consider the case where different elements are pruned before and after quantization. This can only occur if there exists $i, j \in T$ such that $|\mathbf{w}_i| < |\mathbf{w}_j|$, but $\left| \overline{\mathbf{Q}} \left( \mathbf{w} \right)_i \right| = \left| \overline{\mathbf{Q}} \left( \mathbf{w} \right)_j \right|$. Furthermore, suppose the $j$th element is pruned after quantization but not pruned before quantization, while the $i$th element is pruned before quantization and not after. Then, we have that

$$\left| \mathbf{w}_i - \overline{\mathbf{S}} \left( \overline{\mathbf{Q}} \left( \mathbf{w} \right) \right)_i \right| = \left| \mathbf{w}_i - \overline{\mathbf{Q}} \left( \mathbf{w} \right)_i \right|$$
$$\left| \mathbf{w}_j - \overline{\mathbf{S}} \left( \overline{\mathbf{Q}} \left( \mathbf{w} \right) \right)_j \right| = |\mathbf{w}_j|$$

Our goal will be to show that

$$\left| \mathbf{w}_i - \overline{\mathbf{Q}} \left( \overline{\mathbf{S}} \left( \mathbf{w} \right) \right)_i \right|^p + \left| \mathbf{w}_j - \overline{\mathbf{Q}} \left( \overline{\mathbf{S}} \left( \mathbf{w} \right) \right)_j \right|^p \leq \left| \mathbf{w}_i - \overline{\mathbf{S}} \left( \overline{\mathbf{Q}} \left( \mathbf{w} \right) \right)_i \right|^p + \left| \mathbf{w}_j - \overline{\mathbf{S}} \left( \overline{\mathbf{Q}} \left( \mathbf{w} \right) \right)_j \right|^p .$$

Then, by the inductive hypothesis, we will have that

$$\sum_{i \in T \setminus \{i,j\}} \left| \mathbf{w}_i - \overline{\mathbf{Q}} \left( \overline{\mathbf{S}} \left( \mathbf{w} \right) \right)_i \right|^p \leq \sum_{i \in T \setminus \{i,j\}} \left| \mathbf{w}_i - \overline{\mathbf{S}} \left( \overline{\mathbf{Q}} \left( \mathbf{w} \right) \right)_i \right|^p .$$

Thus, we will have that

$$\sum_{i \in T} \left| \mathbf{w}_i - \overline{\mathbf{Q}} \left( \overline{\mathbf{S}} \left( \mathbf{w} \right) \right)_i \right|^p \leq \sum_{i \in T} \left| \mathbf{w}_i - \overline{\mathbf{S}} \left( \overline{\mathbf{Q}} \left( \mathbf{w} \right) \right)_i \right|^p .$$

We assume that $\mathbf{w}_i$ was pruned before quantization and not after. Thus, we have the desired inequality is

$$|\mathbf{w}_i|^p + \left| \mathbf{w}_j - \overline{\mathbf{Q}} \left( \mathbf{w} \right)_j \right|^p \leq \left| \mathbf{w}_i - \overline{\mathbf{Q}} \left( \mathbf{w} \right)_i \right|^p + |\mathbf{w}_j|^p .$$

First, suppose that $\mathbf{w}_i$ and $\mathbf{w}_j$ are opposite signs. Construct $\tilde{\mathbf{w}}$ as described in Lemma 8. We have that $\tilde{\mathbf{w}}_i = -\mathbf{w}_i$ and $\tilde{\mathbf{w}}_j = \mathbf{w}_j$. Furthermore, by Lemma 8, we have that $\overline{\mathbf{Q}} \left( \tilde{\mathbf{w}} \right)_i = -\overline{\mathbf{Q}} \left( \mathbf{w} \right)_i$ and $\overline{\mathbf{Q}} \left( \tilde{\mathbf{w}} \right)_j = \overline{\mathbf{Q}} \left( \mathbf{w} \right)_j$. Thus, we have that $|\tilde{\mathbf{w}}_i| = |\mathbf{w}_i|$, $|\tilde{\mathbf{w}}_j| = |\mathbf{w}_j|$, $\left| \tilde{\mathbf{w}}_i - \overline{\mathbf{Q}} \left( \tilde{\mathbf{w}} \right)_i \right| = \left| \mathbf{w}_i \overline{\mathbf{Q}} \left( \mathbf{w} \right)_i \right|$, $\left| \tilde{\mathbf{w}}_j - \overline{\mathbf{Q}} \left( \tilde{\mathbf{w}} \right)_j \right| = \left| \mathbf{w}_j - \overline{\mathbf{Q}} \left( \mathbf{w} \right)_j \right|$. Therefore, we can equivalently show that

$$|\tilde{\mathbf{w}}_i|^p + \left| \tilde{\mathbf{w}}_j - \overline{\mathbf{Q}} \left( \tilde{\mathbf{w}} \right)_j \right|^p \leq \left| \tilde{\mathbf{w}}_i - \overline{\mathbf{Q}} \left( \tilde{\mathbf{w}} \right)_i \right|^p + |\tilde{\mathbf{w}}_j|^p .$$

Thus, it suffices to consider when $\mathbf{w}_i$ and $\mathbf{w}_j$ share the same sign.

Assume that $\mathbf{w}_i$ and $\mathbf{w}_j$ are the same sign. Without loss of generality, assume that $\mathbf{w}_i, \mathbf{w}_j \geq 0$. Then, by Theorem 5 and Lemma 7, we have that $\overline{\mathbf{Q}}(\mathbf{w})_i = \overline{\mathbf{Q}}(\mathbf{w})_j \geq 0$. Then, we consider the following three cases: $|\mathbf{w}_i| \leq |\overline{\mathbf{Q}}(\mathbf{w})_i| \leq |\mathbf{w}_j|$, $|\mathbf{w}_i| \leq |\mathbf{w}_j| \leq |\overline{\mathbf{Q}}(\mathbf{w})_i|$, and $|\overline{\mathbf{Q}}(\mathbf{w})_i| \leq |\mathbf{w}_i| \leq |\mathbf{w}_j|$.

**Case 1**: $|\mathbf{w}_i| \leq |\overline{\mathbf{Q}}(\mathbf{w})_i| \leq |\mathbf{w}_j|$. We want to show that

$$|\mathbf{w}_i|^p + \left|\mathbf{w}_j - \overline{\mathbf{Q}}(\mathbf{w})_i\right|^p \leq \left|\mathbf{w}_i - \overline{\mathbf{Q}}(\mathbf{w})_i\right|^p + |\mathbf{w}_j|^p.$$

We claim that $|\mathbf{w}_i|^p + \left|\mathbf{w}_j - \overline{\mathbf{Q}}(\mathbf{w})_i\right|^p$ is decreasing on $\overline{\mathbf{Q}}(\mathbf{w})_i$ assuming that $\mathbf{w}_i \leq \overline{\mathbf{Q}}(\mathbf{w})_i \leq \mathbf{w}_j$. We have that $\left|\mathbf{w}_j - \overline{\mathbf{Q}}(\mathbf{w})_i\right| = \mathbf{w}_j - \overline{\mathbf{Q}}(\mathbf{w})_i$. Thus, we have that

$$|\mathbf{w}_i|^p + \left|\mathbf{w}_j - \overline{\mathbf{Q}}(\mathbf{w})_i\right|^p = |\mathbf{w}_i|^p + \left(\mathbf{w}_j - \overline{\mathbf{Q}}(\mathbf{w})_i\right)^p,$$

which is clearly decreasing on $\overline{\mathbf{Q}}(\mathbf{w})_i$. Thus, we have that

$$|\mathbf{w}_i|^p + \left|\mathbf{w}_j - \overline{\mathbf{Q}}(\mathbf{w})_i\right|^p \leq |\mathbf{w}_i|^p + |\mathbf{w}_j - \mathbf{w}_i|^p.$$

Then, we claim that $|\mathbf{w}_i|^p + |\mathbf{w}_j - \mathbf{w}_i|^p \leq |\mathbf{w}_j|^p$. We will equivalently show that

$$|\mathbf{w}_i|^p \leq |\mathbf{w}_j|^p - |\mathbf{w}_j - \mathbf{w}_i|^p.$$

We have that

$$|\mathbf{w}_i|^p = \mathbf{w}_i^p = \int_0^{\mathbf{w}_i} px^{p-1} dx$$

and

$$|\mathbf{w}_j|^p - |\mathbf{w}_j - \mathbf{w}_i|^p = \int_0^{\mathbf{w}_i} p\left(\mathbf{w}_j - \mathbf{w}_i + x\right)^{p-1} dx.$$

We have that $\mathbf{w}_i < \mathbf{w}_j$, so $px^{p-1} < p\left(\mathbf{w}_j - \mathbf{w}_i + x\right)^{p-1}$ for $x \in [0, \mathbf{w}_i]$. Thus, we have that

$$\int_0^{\mathbf{w}_i} px^{p-1} dx \leq \int_0^{\mathbf{w}_i} p\left(\mathbf{w}_j - \mathbf{w}_i + x\right)^{p-1} dx$$

and

$$|\mathbf{w}_i|^p \leq |\mathbf{w}_j|^p - |\mathbf{w}_j - \mathbf{w}_i|^p.$$

Thus, we have that

$$|\mathbf{w}_i|^p + \left|\mathbf{w}_j - \overline{\mathbf{Q}}(\mathbf{w})_i\right|^p \leq |\mathbf{w}_i|^p + |\mathbf{w}_j - \mathbf{w}_i|^p \leq |\mathbf{w}_j|^p.$$

Finally, we have that

$$|\mathbf{w}_j|^p \leq |\mathbf{w}_j|^p + \left|\mathbf{w}_i - \overline{\mathbf{Q}}(\mathbf{w})_i\right|^p,$$

since $\left|\mathbf{w}_i - \overline{\mathbf{Q}}(\mathbf{w})_i\right| \geq 0$. Therefore, we have proved the desired inequality.

**Case 2**: $|\mathbf{w}_i| \leq |\mathbf{w}_j| \leq |\overline{\mathbf{Q}}(\mathbf{w})_i|$. We want to show that

$$|\mathbf{w}_i|^p + \left|\mathbf{w}_j - \overline{\mathbf{Q}}(\mathbf{w})_i\right|^p \leq \left|\mathbf{w}_i - \overline{\mathbf{Q}}(\mathbf{w})_i\right|^p + |\mathbf{w}_j|^p.$$

Rearranging, we have that

$$\left|\mathbf{w}_i - \overline{\mathbf{Q}}(\mathbf{w})_i\right|^p - \left|\mathbf{w}_j - \overline{\mathbf{Q}}(\mathbf{w})_i\right|^p \geq |\mathbf{w}_i|^p - |\mathbf{w}_j|^p.$$

First, we claim that $\left|\mathbf{w}_i - \overline{\mathbf{Q}}(\mathbf{w})_i\right|^p - \left|\mathbf{w}_j - \overline{\mathbf{Q}}(\mathbf{w})_i\right|^p$ is increasing in $\overline{\mathbf{Q}}(\mathbf{w})_i$. Since $|\mathbf{w}_i| \leq |\mathbf{w}_j| \leq |\overline{\mathbf{Q}}(\mathbf{w})_i|$ and we are assuming that $\mathbf{w}_i, \mathbf{w}_j, \overline{\mathbf{Q}}(\mathbf{w})_i \geq 0$, then we have that

$$\left|\mathbf{w}_j - \overline{\mathbf{Q}}(\mathbf{w})_i\right| = \overline{\mathbf{Q}}(\mathbf{w})_i - \mathbf{w}_j$$

and

$$\left|\mathbf{w}_i - \overline{\mathbf{Q}}(\mathbf{w})_i\right| = \overline{\mathbf{Q}}(\mathbf{w})_i - \mathbf{w}_i.$$

Thus, it suffices to show that $\left(\overline{\mathbf{Q}}(\mathbf{w})_i - \mathbf{w}_i\right)^p - \left(\overline{\mathbf{Q}}(\mathbf{w})_i - \mathbf{w}_j\right)^p$ is increasing in $\overline{\mathbf{Q}}(\mathbf{w})_i$. We have that

$$\left(\overline{\mathbf{Q}}(\mathbf{w})_i - \mathbf{w}_i\right)^p - \left(\overline{\mathbf{Q}}(\mathbf{w})_i - \mathbf{w}_j\right)^p = \int_{-\mathbf{w}_j}^{-\mathbf{w}_i} p\left(\overline{\mathbf{Q}}(\mathbf{w})_i + x\right)^{p-1} dx.$$

We have that $p \left( \overline{\mathbf{Q}} \left( \mathbf{w} \right)_i + x \right)^{p-1}$ is increasing in $\overline{\mathbf{Q}} \left( \mathbf{w} \right)_i$, and $-\mathbf{w}_j < -\mathbf{w}_i$ by assumption. Therefore, $\left( \overline{\mathbf{Q}} \left( \mathbf{w} \right)_i - \mathbf{w}_i \right)^p - \left( \overline{\mathbf{Q}} \left( \mathbf{w} \right)_i - \mathbf{w}_j \right)^p$ is increasing in $\overline{\mathbf{Q}} \left( \mathbf{w} \right)_i$. Therefore, we have that

$$\left| \mathbf{w}_i - \overline{\mathbf{Q}} \left( \mathbf{w} \right)_i \right|^p - \left| \mathbf{w}_j - \overline{\mathbf{Q}} \left( \mathbf{w} \right)_i \right|^p \geq \left| \mathbf{w}_i - \mathbf{w}_j \right|^p - \left| \mathbf{w}_j - \mathbf{w}_j \right|^p = \left| \mathbf{w}_i - \mathbf{w}_j \right|^p > 0.$$

Furthermore, we have that $\left| \mathbf{w}_i \right|^p - \left| \mathbf{w}_j \right|^p < 0$, since $0 \leq \mathbf{w}_i < \mathbf{w}_j$. Therefore, the desired inequality is shown.

**Case 3:** $\left| \overline{\mathbf{Q}} \left( \mathbf{w} \right)_i \right| \leq \left| \mathbf{w}_i \right| \leq \left| \mathbf{w}_j \right|$. Again, we want to show that

$$\left| \mathbf{w}_i \right|^p + \left| \mathbf{w}_j - \overline{\mathbf{Q}} \left( \mathbf{w} \right)_i \right|^p \leq \left| \mathbf{w}_i - \overline{\mathbf{Q}} \left( \mathbf{w} \right)_i \right|^p + \left| \mathbf{w}_j \right|^p.$$

Rearranging, we have that

$$\left| \mathbf{w}_i \right|^p - \left| \mathbf{w}_i - \overline{\mathbf{Q}} \left( \mathbf{w} \right)_i \right|^p \leq \left| \mathbf{w}_j \right|^p - \left| \mathbf{w}_j - \overline{\mathbf{Q}} \left( \mathbf{w} \right)_i \right|^p.$$

Then, we have that

$$\left| \mathbf{w}_i \right|^p - \left| \mathbf{w}_i - \overline{\mathbf{Q}} \left( \mathbf{w} \right)_i \right|^p = \int_{-\overline{\mathbf{Q}}(\mathbf{w})_i}^0 p \left( \mathbf{w}_j + x \right)^{p-1} dx$$

and

$$\left| \mathbf{w}_j \right|^p - \left| \mathbf{w}_j - \overline{\mathbf{Q}} \left( \mathbf{w} \right)_i \right|^p = \int_{-\overline{\mathbf{Q}}(\mathbf{w})_i}^0 p \left( \mathbf{w}_i + x \right)^{p-1} dx.$$

We have that $-\overline{\mathbf{Q}} \left( \mathbf{w} \right)_i < 0$ and $\mathbf{w}_i < \mathbf{w}_j$. Thus,

$$\int_{-\overline{\mathbf{Q}}(\mathbf{w})_i}^0 p \left( \mathbf{w}_j + x \right)^{p-1} dx < \int_{-\overline{\mathbf{Q}}(\mathbf{w})_i}^0 p \left( \mathbf{w}_i + x \right)^{p-1} dx.$$

Therefore, we have the inequality as desired.

This proves the statement for finite $p$. Note that the $L_\infty$ norm is the limit of the $L_p$ norm as $p \to \infty$. Thus, using the proven inequality and taking the limit as $p \to \infty$, this proves the corresponding result for the $L_\infty$ norm. $\qquad \square$

## D  COUNTEREXAMPLE FOR NON-NAIVE QUANTIZATION SCHEMES

Theorem 1 no longer holds under non-naive quantization schemes. We provide a counterexample. Let

$$\mathbf{w} = \begin{bmatrix} 0.1 \\ 0.9 \end{bmatrix}.$$

Let $\mathcal{Q}_i = \frac{3}{2} \mathbb{Z}$ for all $i \in [d]$. Furthermore, let $\Phi, D_i$ be defined the same way as in a naive symmetric max-scaled block-wise quantization scheme. However, we modify $Q_i$ slightly. In particular, like AdaRound, we choose a random direction to round. Thus, we have that

$$\Phi \left( \mathbf{w} \right) = [0.9].$$

Then, suppose that

$$\mathbf{Q} \left( \mathbf{w} \right) = \begin{bmatrix} 0 \\ \frac{3}{2} \end{bmatrix}.$$

Then, we have that

$$\overline{\mathbf{Q}} \left( \mathbf{w} \right) = \begin{bmatrix} 0 \\ 1.35 \end{bmatrix}.$$

Now, suppose $\overline{\mathbf{S}}$ is a $50\%$ magnitude-based block-wise sparsification scheme. Then, we have that

$$\overline{\mathbf{S}} \left( \mathbf{w} \right) = \begin{bmatrix} 0 \\ 1.8 \end{bmatrix}.$$

Furthermore, let

$$\mathbf{Q} \left( \overline{\mathbf{S}} \left( \mathbf{w} \right) \right) = \overline{\mathbf{Q}} \left( \begin{bmatrix} 0 \\ 1.8 \end{bmatrix} \right) = \begin{bmatrix} 0 \\ 0 \end{bmatrix}.$$

Here, we assume that we randomly choose to round down from $0.9$ to $0$, instead of round up to $1.35$.

Then, we have that

$$\left\| \mathbf{w} - \overline{\mathbf{Q}}\left( \overline{\mathbf{S}}\left( \mathbf{w} \right)\right)\right\|_1 = \left\| \begin{bmatrix} 0.1 \\ 0.9 \end{bmatrix} \right\|_1 = 1.$$

We also have that

$$\left\| \mathbf{w} - \overline{\mathbf{S}}\left( \overline{\mathbf{Q}}\left( \mathbf{w} \right)\right)\right\|_1 = \left\| \begin{bmatrix} 0.1 \\ -0.45 \end{bmatrix} \right\|_1 = 0.55,$$

since

$$\overline{\mathbf{S}}\left( \overline{\mathbf{Q}}\left( \mathbf{w} \right)\right) = \begin{bmatrix} 0 \\ 1.35 \end{bmatrix}.$$

Therefore, it is also not necessarily true that

$$\left\| \mathbf{w} - \overline{\mathbf{Q}}\left( \overline{\mathbf{S}}\left( \mathbf{w} \right)\right)\right\|_1 \leq \left\| \mathbf{w} - \overline{\mathbf{S}}\left( \overline{\mathbf{Q}}\left( \mathbf{w} \right)\right)\right\|_1.$$

# E    PROOF OF THEOREM 3

*Proof.* We use induction on $n$, defined as the number of entries that are sparsified in exactly one of $\mathbf{Q} \to \mathbf{S}$ and $\mathbf{S} \to \mathbf{Q}$. The base case, $n = 0$, is trivial, since the weights will be the same. Otherwise, consider any indices $i, j$ that are pruned differently by the two procedures. Since we only care about the magnitudes of the weights, like in the proof for Theorem 1, assume that $\mathbf{w}_i \geq 0, \mathbf{w}_j \geq 0$. Without loss of generality, we further assume that $\mathbf{w}_i \leq \mathbf{w}_j$. We know that $\overline{\mathbf{Q}}(\mathbf{w})_i \leq \overline{\mathbf{Q}}(\mathbf{w})_j$ by Theorem 5. Let $h_{ii}$ be the $i$th diagonal element of $\mathbf{H}_L\left( \mathbf{w} \right)$.

Note that we cannot have either of $\mathbf{w}_i$ equal to zero, since the only way $\mathbf{Q} \to \mathbf{S}$ and $\mathbf{S} \to \mathbf{Q}$ could prune different weights would be for $\overline{\mathbf{Q}}(\mathbf{w}_j) = 0$, but this would imply that elements $i$ and $j$ of $\mathbf{w}_{\mathbf{Q} \to \mathbf{S}}$ and $\mathbf{w}_{\mathbf{S} \to \mathbf{Q}}$ are equal, which is a contradiction. Thus, we can restrict without loss of generality to the case $\mathbf{w}_i > 0, \mathbf{w}_j > 0$.

Then there are two cases.

**Case 1:** $h_{ii}\mathbf{w}_i^2 \geq h_{jj}\mathbf{w}_j^2$. Since the two procedures prune different weights, this implies $h_{ii}\overline{\mathbf{Q}}(\mathbf{w})_i^2 < h_{jj}\overline{\mathbf{Q}}(\mathbf{w})_j^2$. This implies $\mathbf{S} \to \mathbf{Q}$ will prune $\mathbf{w}_j$, while $\mathbf{Q} \to \mathbf{S}$ will prune $\mathbf{w}_i$. We wish to show

$$L\left( \hat{\mathbf{w}}_{\mathbf{S} \to \mathbf{Q}} \right) \leq L\left( \hat{\mathbf{w}}_{\mathbf{Q} \to \mathbf{S}} \right) + \mathcal{O}(\|\hat{\mathbf{w}}_{\mathbf{Q} \to \mathbf{S}} - \mathbf{w}\|^3),$$

Note that when using a second-order Taylor expansion, $\mathcal{O}(\|\hat{\mathbf{w}}_{\mathbf{S} \to \mathbf{Q}} - \mathbf{w}\|^3) \subseteq \mathcal{O}(\|\hat{\mathbf{w}}_{\mathbf{Q} \to \mathbf{S}} - \mathbf{w}\|^3)$ by Theorem 1. Then using the inductive hypothesis, it suffices to show the terms indexed by $i$ and $j$ in the loss satisfy the desired inequality, so it suffices to show that

$$h_{ii}(\mathbf{w}_i - \overline{\mathbf{Q}}(\mathbf{w})_i)^2 + h_{jj}\mathbf{w}_j^2 \leq h_{ii}\mathbf{w}_i^2 + h_{jj}(\mathbf{w}_j - \overline{\mathbf{Q}}(\mathbf{w})_j)^2$$

which can be written as

$$2(h_{jj}\mathbf{w}_j\overline{\mathbf{Q}}(\mathbf{w})_j - h_{ii}\mathbf{w}_i\overline{\mathbf{Q}}(\mathbf{w})_i) + (h_{ii}\overline{\mathbf{Q}}(\mathbf{w})_i^2 - h_{jj}\overline{\mathbf{Q}}(\mathbf{w})_j^2) \leq 0.$$

We already know the second term $h_{ii}\overline{\mathbf{Q}}(\mathbf{w})_i^2 - h_{jj}\overline{\mathbf{Q}}(\mathbf{w})_j^2 < 0$. Then, note that we have

$$\frac{\mathbf{w}_j}{\mathbf{w}_i} \leq \frac{\sqrt{h_{ii}}}{\sqrt{h_{jj}}}.$$

For a quantization grid of step size $\delta$ with $0 < \delta < 2\mathbf{w}_i$,

$$\frac{\overline{\mathbf{Q}}(\mathbf{w})_j}{\overline{\mathbf{Q}}(\mathbf{w})_i} \leq \frac{\mathbf{w}_j + \delta/2}{\mathbf{w}_i - \delta/2} \leq \frac{\mathbf{w}_j}{\mathbf{w}_i} \cdot \frac{1 + \frac{\delta}{2\mathbf{w}_j}}{1 - \frac{\delta}{2\mathbf{w}_i}} \leq \frac{\sqrt{h_{ii}}}{\sqrt{h_{jj}}} \cdot \frac{1 + \frac{\delta}{2\mathbf{w}_j}}{1 - \frac{\delta}{2\mathbf{w}_i}}.$$

Let $c_1(\delta) = (1 + \frac{\delta}{2\mathbf{w}_j})/(1 - \frac{\delta}{2\mathbf{w}_i})$, where we know that as $\lim_{\delta \to 0} c_1(\delta) = 1$. Combining the two inequalities, this implies that

$$\mathbf{w}_j\overline{\mathbf{Q}}(\mathbf{w})_j \leq \frac{h_{ii}}{h_{jj}}c_1(\delta)\mathbf{w}_i\overline{\mathbf{Q}}(\mathbf{w})_i,$$

so that

$$2(h_{jj}\mathbf{w}_j\overline{\mathbf{Q}}(\mathbf{w})_j - h_{ii}\mathbf{w}_i\overline{\mathbf{Q}}(\mathbf{w})_i) \le 2h_{ii}\mathbf{w}_i\overline{\mathbf{Q}}(\mathbf{w})_i(c_1(\delta) - 1),$$

which approaches zero as $\delta \to 0$. Since we know that $h_{ii}\overline{\mathbf{Q}}(\mathbf{w})_i^2 - h_{jj}\overline{\mathbf{Q}}(\mathbf{w})_j^2 < 0$, this implies there exists some $\varepsilon_{i,j,1} > 0$ such that $\delta < \varepsilon_{i,j,1}$ and $\sqrt{h_{ii}}\mathbf{w}_i \ge \sqrt{h_{jj}}\mathbf{w}_j$ imply $2(h_{jj}\mathbf{w}_j\overline{\mathbf{Q}}(\mathbf{w})_j - h_{ii}\mathbf{w}_i\overline{\mathbf{Q}}(\mathbf{w})_i) + (h_{ii}\overline{\mathbf{Q}}(\mathbf{w})_i^2 - h_{jj}\overline{\mathbf{Q}}(\mathbf{w})_j^2) \le 0$.

**Case 2:** $h_{ii}\mathbf{w}_i^2 \le h_{jj}\mathbf{w}_j^2$. Since the two procedures prune different weights, this implies $h_{ii}\overline{\mathbf{Q}}(\mathbf{w})_i^2 \ge h_{jj}\overline{\mathbf{Q}}(\mathbf{w})_j^2$. This implies $\mathbf{S} \to \mathbf{Q}$ will prune $\mathbf{w}_i$, while $\mathbf{Q} \to \mathbf{S}$ will prune $\mathbf{w}_j$. Hence, since only two terms in the loss function are differing, we can perform similar steps to case 1. By the inductive hypothesis and a second-order Taylor expansion, it suffices to show that

$$h_{ii}\mathbf{w}_i^2 + h_{jj}(\mathbf{w}_j - \overline{\mathbf{Q}}(\mathbf{w})_j)^2 \le h_{ii}(\mathbf{w}_i - \overline{\mathbf{Q}}(\mathbf{w})_i)^2 + h_{jj}\mathbf{w}_j^2$$

which reduces to

$$2(h_{jj}\mathbf{w}_j\overline{\mathbf{Q}}(\mathbf{w})_j - h_{ii}\mathbf{w}_i\overline{\mathbf{Q}}(\mathbf{w})_i) + (h_{ii}\overline{\mathbf{Q}}(\mathbf{w})_i^2 - h_{jj}\overline{\mathbf{Q}}(\mathbf{w})_j^2) \ge 0$$

We already know the second term $h_{ii}\overline{\mathbf{Q}}(\mathbf{w})_i^2 - h_{jj}\overline{\mathbf{Q}}(\mathbf{w})_j^2 \ge 0$. For a quantization grid of step size $\delta$, when $0 < \delta < 2\mathbf{w}_j$,

$$\frac{\overline{\mathbf{Q}}(\mathbf{w})_j}{\overline{\mathbf{Q}}(\mathbf{w})_i} \ge \frac{\mathbf{w}_j - \delta/2}{\mathbf{w}_i + \delta/2} = \frac{\mathbf{w}_j}{\mathbf{w}_i} \cdot \frac{1 - \frac{\delta}{2\mathbf{w}_j}}{1 + \frac{\delta}{2\mathbf{w}_i}} \ge \frac{\sqrt{h_{ii}}}{\sqrt{h_{jj}}} \cdot \frac{1 - \frac{\delta}{2\mathbf{w}_j}}{1 + \frac{\delta}{2\mathbf{w}_i}},$$

where we define $c_2(\delta) = (1 - \frac{\delta}{2\mathbf{w}_j})/(1 + \frac{\delta}{2\mathbf{w}_i})$. Combining this inequality with the fact that $h_{jj}\mathbf{w}_j^2 \ge h_{ii}\mathbf{w}_i^2$, we see that

$$\mathbf{w}_j\overline{\mathbf{Q}}(\mathbf{w})_j \ge \frac{h_{ii}}{h_{jj}}c_2(\delta)\mathbf{w}_i\overline{\mathbf{Q}}(\mathbf{w})_i,$$

so that

$$2(h_{jj}\mathbf{w}_j\overline{\mathbf{Q}}(\mathbf{w})_j - h_{ii}\mathbf{w}_i\overline{\mathbf{Q}}(\mathbf{w})_i) \ge 2h_{ii}\mathbf{w}_i\overline{\mathbf{Q}}(\mathbf{w})_i(c_w(\delta) - 1),$$

which approaches zero as $\delta \to 0$. Thus, this shows that this implies there exists some $\varepsilon_{i,j,2} > 0$ such that $\delta < \varepsilon_{i,j,2}$ and $\sqrt{h_{ii}}\mathbf{w}_i \le \sqrt{h_{jj}}\mathbf{w}_j$ imply $2(h_{jj}\mathbf{w}_j\overline{\mathbf{Q}}(\mathbf{w})_j - h_{ii}\mathbf{w}_i\overline{\mathbf{Q}}(\mathbf{w})_i) + (h_{ii}\overline{\mathbf{Q}}(\mathbf{w})_i^2 - h_{jj}\overline{\mathbf{Q}}(\mathbf{w})_j^2) \ge 0$.

This completes the induction, so we have shown that

$$L(\hat{\mathbf{w}}_{\mathbf{S}\to\mathbf{Q}}) \le L(\hat{\mathbf{w}}_{\mathbf{Q}\to\mathbf{S}}) + \mathcal{O}(\|\hat{\mathbf{w}}_{\mathbf{Q}\to\mathbf{S}} - \mathbf{w}\|^3).$$

Since there are finitely many inductive steps, this bound holds for some $\varepsilon > 0$. For example, we can conservatively take $\varepsilon = \min\{\varepsilon_{i,j,k}\}_{k\in\{1,2\}, i\ne j}$. $\qquad\square$

# F    PROOF OF THEOREM 4

*Proof.* Suppose we have a quantization scheme followed by a sparsification scheme that introduces a $\Delta\mathbf{w}_{\mathbf{Q}} + \Delta\mathbf{w}_{\mathbf{S}|\mathbf{Q}}$ change to the weights. Then, we had the following, assuming that $\mathbf{w}$ was a local optimum.

$$L(\mathbf{w} + \Delta\mathbf{w}_{\mathbf{Q}} + \Delta\mathbf{w}_{\mathbf{S}|\mathbf{Q}}) - L(\mathbf{w})$$
$$= \frac{1}{2}(\Delta\mathbf{w}_{\mathbf{Q}} + \Delta\mathbf{w}_{\mathbf{S}|\mathbf{Q}})^{\mathsf{T}}\mathbf{H}_L(\mathbf{w})(\Delta\mathbf{w}_{\mathbf{Q}} + \Delta\mathbf{w}_{\mathbf{S}|\mathbf{Q}}) + \mathcal{O}(\|\Delta\mathbf{w}_{\mathbf{Q}} + \Delta\mathbf{w}_{\mathbf{S}|\mathbf{Q}}\|^3)$$

Similarly, we assume that we have a sparsification scheme followed by a quantization scheme that introduces $\Delta\mathbf{w}_{\mathbf{S}} + \Delta\mathbf{w}_{\mathbf{Q}|\mathbf{S}}$ change to the weights. Then, we had the following, assuming that $\mathbf{w}$ was a local optimum.

$$L(\mathbf{w} + \Delta\mathbf{w}_{\mathbf{S}} + \Delta\mathbf{w}_{\mathbf{Q}|\mathbf{S}}) - L(\mathbf{w})$$
$$= \frac{1}{2}(\Delta\mathbf{w}_{\mathbf{S}} + \Delta\mathbf{w}_{\mathbf{Q}|\mathbf{S}})^{\mathsf{T}}\mathbf{H}_L(\mathbf{w})(\Delta\mathbf{w}_{\mathbf{S}} + \Delta\mathbf{w}_{\mathbf{Q}|\mathbf{S}}) + \mathcal{O}(\|\Delta\mathbf{w}_{\mathbf{S}} + \Delta\mathbf{w}_{\mathbf{Q}|\mathbf{S}}\|^3)$$

Using the fact that $\varepsilon_{\mathbf{S}} = \Delta\mathbf{w}_{\mathbf{S}|\mathbf{Q}} - \Delta\mathbf{w}_{\mathbf{S}}$, we get the following for the change in loss under $\mathbf{Q} \to \mathbf{S}$.

$$
L\left(\mathbf{w} + \Delta\mathbf{w}_{\mathbf{Q}} + \Delta\mathbf{w}_{\mathbf{S}|\mathbf{Q}}\right) - L\left(\mathbf{w}\right)
$$
$$
= \frac{1}{2}\left(\Delta\mathbf{w}_{\mathbf{Q}} + \Delta\mathbf{w}_{\mathbf{S}} + \varepsilon_{\mathbf{S}}\right)^{\mathsf{T}} \mathbf{H}_L\left(\mathbf{w}\right)\left(\Delta\mathbf{w}_{\mathbf{Q}} + \Delta\mathbf{w}_{\mathbf{S}} + \varepsilon_{\mathbf{S}}\right) + \mathcal{O}\left(\|\Delta\mathbf{w}_{\mathbf{Q}} + \Delta\mathbf{w}_{\mathbf{S}|\mathbf{Q}}\|^3\right)
$$
$$
= \frac{1}{2}\left(\Delta\mathbf{w}_{\mathbf{Q}} + \Delta\mathbf{w}_{\mathbf{S}}\right)^{\mathsf{T}} \mathbf{H}_L\left(\mathbf{w}\right)\left(\Delta\mathbf{w}_{\mathbf{Q}} + \Delta\mathbf{w}_{\mathbf{S}}\right)
$$
$$
+ \varepsilon_{\mathbf{S}}^{\mathsf{T}}\mathbf{H}_L\left(\mathbf{w}\right)\left(\Delta\mathbf{w}_{\mathbf{Q}} + \Delta\mathbf{w}_{\mathbf{S}}\right)
$$
$$
+ \frac{1}{2}\varepsilon_{\mathbf{S}}^{\mathsf{T}}\mathbf{H}_L\left(\mathbf{w}\right)\varepsilon_{\mathbf{S}} + \mathcal{O}\left(\|\Delta\mathbf{w}_{\mathbf{Q}} + \Delta\mathbf{w}_{\mathbf{S}|\mathbf{Q}}\|^3\right)
$$

We have the following for $\mathbf{S} \to \mathbf{Q}$ under the assumption that $\Delta\mathbf{w}_{\mathbf{Q}|\mathbf{S}} = \Delta\mathbf{w}_{\mathbf{Q}}$.

$$
L\left(\mathbf{w} + \Delta\mathbf{w}_{\mathbf{S}} + \Delta\mathbf{w}_{\mathbf{Q}|\mathbf{S}}\right) - L\left(\mathbf{w}\right)
$$
$$
= \frac{1}{2}\left(\Delta\mathbf{w}_{\mathbf{S}} + \Delta\mathbf{w}_{\mathbf{Q}}\right)^{\mathsf{T}} \mathbf{H}_L\left(\mathbf{w}\right)\left(\Delta\mathbf{w}_{\mathbf{S}} + \Delta\mathbf{w}_{\mathbf{Q}}\right) + \mathcal{O}\left(\|\Delta\mathbf{w}_{\mathbf{S}} + \Delta\mathbf{w}_{\mathbf{Q}|\mathbf{S}}\|^3\right)
$$

Now, we take the difference between the change in loss for $\mathbf{Q} \to \mathbf{S}$ and $\mathbf{S} \to \mathbf{Q}$.

$$
L\left(\mathbf{w} + \Delta\mathbf{w}_{\mathbf{Q}} + \Delta\mathbf{w}_{\mathbf{S}|\mathbf{Q}}\right) - L\left(\mathbf{w} + \Delta\mathbf{w}_{\mathbf{S}} + \Delta\mathbf{w}_{\mathbf{Q}|\mathbf{S}}\right)
$$
$$
= \varepsilon_{\mathbf{S}}^{\mathsf{T}}\mathbf{H}_L\left(\mathbf{w}\right)\left(\Delta\mathbf{w}_{\mathbf{Q}} + \Delta\mathbf{w}_{\mathbf{S}}\right) + \frac{1}{2}\varepsilon_{\mathbf{S}}^{\mathsf{T}}\mathbf{H}_L\left(\mathbf{w}\right)\varepsilon_{\mathbf{S}} + \mathcal{O}\left(\|\Delta\mathbf{w}_{\mathbf{Q}\to\mathbf{S}}\|^3\right) + \mathcal{O}\left(\|\Delta\mathbf{w}_{\mathbf{S}\to\mathbf{Q}}\|^3\right)
$$

$\square$

