# OpenReview forum: "Towards a Unified Theory of Quantization and Sparsity"
_ICLR.cc/2026/Conference — Submitted to ICLR 2026_

### Official Review · Reviewer_9bPN · 2025-10-23

**Soundness:** 2
**Presentation:** 3
**Contribution:** 2
**Rating:** 4
**Confidence:** 4

**Summary:**

The work advances the theoretical understanding of combining sparsity and quantisation for compressing deep learning models. First, the authors generalise previous results on the advantage (w.r.t. parameter reconstruction error) of running sparsification-before-quantisation vs quantisation-before-sparsification. Then, the authors consider the model-level with a diagonal Hessian, equivalent to a $\mathrm{diag}(H)$-weighted squared reconstruction error, showing that (under some conditions) an advantage for sparsification-first. However this is shown not to hold true in the more general case. A short empirical investigation follows, showing that sparsification-first can be better in practice, but that if the sparsification mask is taken from the original weights, then quantisation-first can also work as well.

**Strengths:**

The work is clearly presented and makes a reasonably robust case for the main claims. Particular strengths:

 - Notation is clear and sufficiently well-explained
 - Helpful intuition is given for some proofs, e.g. in Section 4.1
 - Results for QAS are interesting and somewhat surprising to me
 - The idea in section 6.3 of using a short re-training phase to disambiguate weights that are quantised to the same level is interesting

**Weaknesses:**

**Main concern:** Although I appreciate the rigorous derivation, the results in Theorems 1, 2 and 4 seem trivial and unsurprising. T1 since (as explained in 4.1), weights that are quantised to the same level cannot be disambiguated, T2 is easily reduced to T1, as stated and the claim associated with T4 is weak, having dropped the constraints that made it possible to show T3. I found T3 the most interesting, but I don't completely follow the constraints (question below). Combined with the short and small-scale empirical investigation, I do not take the results as sufficiently informative to the community to recommend acceptance.

**Specific concerns and questions:**

1. The scale bound $\delta \leq \epsilon$ isn't entirely clear to me - do we know if this is likely to be satisfied in practice, if scale is derived from a tensor-absmax?
1. Is the definition of $\epsilon_S$ in Theorem 3 (body) correct? I think as stated the following assumption would reduce to $\epsilon_S = 0$, and seems inconsistent with that of the proof in L1134, which would expect $\epsilon_S \coloneqq \Delta w_{Q \rightarrow S} - \Delta w_Q - \Delta w_S$).
1. Block (group) quantisation is common (and indeed seems to be used by default with AWQ in your supplementary scripts), while as I understand it, many of the theoretical results only apply to weights within a single block.
1. Although it is acceptable to have a limited empirical example given the theoretical focus, I think there are some obvious gaps. Although the theoretical results concern (Q, S) = (Naive Max-Scaled, Magnitude) or (Naive Max-Scaled, OBD), the results do not include these settings, always using AWQ for quantisation and not including OBD for sparsity. It would also greatly help to see sparsity-only and quantisation-only results for comparison.
1. In Table 1, despite the observation that for INT4 with 10% Magnitude pruning, the INT4 quantisation step already sets enough weights to zero to make sparsity a no-op, I can't understand why the more accurate INT8 format should under-perform INT4. (Anything that sparsity flushes to zero in the INT8 case would already have been flushed to zero in INT4.)
1. The model scale, sparsity and quantisation settings considered in the experiments are far from the state of the art (e.g. Tseng et al., 2024, Liu et al., 2024, Frantar et al. 2023), making it hard to be confident that the theoretical results can be observed in practice.

**Minor concerns:**

 - The "key insight" of L168, stating "$|w_i - \bar{Q}(\bar{S}(w))_i| = 0$ if $w_i$ is pruned" doesn't seem right, wouldn't it be $= |w_i|$ in this case?
 - While I accept Theorem 1, the implication stated in L185 concerning $L_{\infty}$ seems potentially misleading - isn't equality very highly likely in this case (assuming tensor-wise quantisation scaling), since the same-quantisation-level collision would have to occur for the extreme value?
 - Body section 4.4 references Theorem 5 from the appendix.
 - In Tables 1 and 2, I most wish to compare Sparsity Method & Order for given Precision & Sparsity levels, which would be much easier if the major grouping was on Precision & Sparsity levels and minor grouping on Sparsity Method & Order.

---

_Tseng, A., Sun, Q., Hou, D. and De Sa, C.M., 2024. Qtip: Quantization with trellises and incoherence processing. Advances in Neural Information Processing Systems, 37, pp.59597-59620._

_Liu Z, Zhao C, Fedorov I, Soran B, Choudhary D, Krishnamoorthi R, Chandra V, Tian Y, Blankevoort T. Spinquant: Llm quantization with learned rotations. arXiv preprint arXiv:2405.16406. 2024 May 26._

_Frantar, E. and Alistarh, D., 2023, July. Sparsegpt: Massive language models can be accurately pruned in one-shot. In International conference on machine learning (pp. 10323-10337). PMLR._

**Questions:**

I would appreciate any clarifications/corrections/counter-arguments, especially regarding my main concern and specific concerns above.

---

> ### Author Response · Authors · 2025-11-21
>
> We greatly appreciate your review of the paper. We believe that the concerns and questions you raise are valid, and we will try to address them.
>
> Specific Concerns and Questions:
> 1. We agree that this assumption is hard to verify in practice, but we included it as it was necessary machinery for the proof. However, it could be possible given that the scale is typically derived as the amax divided by the largest representable value. Many lower precision schemes tend to favor greater range.
> 2. You are correct. Sorry about that. It should read $\boldsymbol{\varepsilon}\_{\mathbf{S}} = \left(\Delta \mathbf{w}\_{\mathbf{Q} \to \mathbf{S}} - \Delta \mathbf{w}\_{\mathbf{Q}}\right) - \Delta \mathbf{w}\_{\mathbf{S}}$.
> 3. Yes, it is correct that block quantization is quite common nowadays. However, we think per-tensor scaling is still used, especially for 8-bit datatypes. While we do use AWQ via block quantization, we use AWQ to provide a counterexample and support Theorem 4, which does not make the assumption that the quantization scheme uses a per-tensor scale.
> 4. We understand the concern for lack of empirical data. For Theorems 1, 2 and 3, the goal was to show that $\mathbf{S} \to \mathbf{Q}$ is better than $\mathbf{Q} \to \mathbf{S}$ in some settings, as is the belief in the current literature. There is a decent amount of empirical data from previous papers in the literature that show empirical data for this phenomenon. We simply included these theorems to try to prove the statements mathematically. Thus, we felt we did not need to include new empirical data. We included empirical data as a counterexample for showing that $\mathbf{Q} \to \mathbf{S}$ may outperform $\mathbf{S} \to \mathbf{Q}$ as that has not been previously seen in the literature.
> 5. I think the assumption that anything that sparsity flushes to zero in the INT8 case would already be flushed to zero in INT4 quantization may not be true. Under the different precisions, the per-channel scales may change, since the quantization error and therefore objective function changes. Thus, we may not be able to assume that the order should be preserved between the INT4 and INT8 cases.
> 6. We agree that the scale, sparsity, and quantization settings are far from the state-of-the-art. We wanted to provide a counterexample under a relatively simple and controlled setting to show the existence that we foreshadowed in the previous section, but we understand that we should have made it closer to a more practical setting.
>
> Minor Concerns:
> 1. Yes, we apologize for this typo. We have corrected this. It should read $\left|\mathbf{w}\_i - \overline{\mathbf{S}}\left(\overline{\mathbf{Q}}\left(\mathbf{w}\right)\right)\_{i}\right| = \left|\mathbf{w}\_i - \overline{\mathbf{Q}}\left(\overline{\mathbf{S}}\left(\mathbf{w}\right)\right)\_{i}\right|$.
> 2. Yes, we agree that this statement is misleading. We have corrected this to say that $\mathbf{S} \to \mathbf{Q}$ will be as good as $\mathbf{Q} \to \mathbf{S}$.
> 3. Yes, sorry for the confusion. We have corrected Section 4.4 to refer to Theorem 1.
> 4. We agree that the grouping you propose would have been more ideal. We have made such changes.
>
> Thank you once again for your careful review!

---

> > ### Comment · Reviewer_9bPN · 2025-11-21
> >
> > Thank you for your comprehensive responses, which are helpful. I particularly appreciate your comment on (5) regarding the effect of quantisation error on the per-channel scales, which is a reasonable explanation for the somewhat counter-intuitive result.

---

### Official Review · Reviewer_Sg9T · 2025-10-31

**Soundness:** 2
**Presentation:** 2
**Contribution:** 1
**Rating:** 2
**Confidence:** 4

**Summary:**

The paper provides a stronger theoretical result that $S \rightarrow Q$ (sparsity before quantization) is optimal over $Q \rightarrow S$ on a tensor level, and the same result on the model level under some assumptions. To mitigate error in $Q \rightarrow S$ setup, the authors propose using the unquantized weights for calculating sparsity mask, which they name Quantisation-Aware Sparsification (QAS). Then they demonstrate on OPT-125M model that $Q \rightarrow S$ can perform on par with  $S \rightarrow Q$ with QAS.

**Strengths:**

1. The paper tackles an open practical question of interaction between quantization and sparsity and highlights the necessity of codesigning the hybrid compression formats.
2. This work proves on a tensor level for any $L_p$ norm that sparsity before quantization minimizes the loss.

**Weaknesses:**

1. Limited novelty. The work closely follows Harma et al. (2025), in particular their claim on $S \rightarrow Q$ being optimal over $Q \rightarrow S$ on a tensor level and empirical validation on a model level. Harma et al. proves the statement of Theorem 1 for $p=1$, and the authors of this work only extend it to arbitrary $p \geq 1$ without gaining new insight.
2. Too strong assumptions: Theorems 2 and 3 assume Hessian to be an identity matrix, and a diagonal matrix, respectively. The condition $\Delta w_{Q|S} = \Delta w_Q$ is particularly restrictive, it basically means the sparsity only affects the weights that would be quantized to the same value as 0, $Q(S(x)) = Q(0) = Q(x)$. This practically renders sparsity unnecessary in this setup.
3. The paper lacks contribution and insight. Proposed Quantisation-Aware Sparsification effectively makes element-wise quantization into quantization applied after sparsification. The paper offers no explanation on how would the training dynamics with QAS change for more complex quantization / sparsity schemes.
4. Limited validation: the authors conduct experiments using a single model instance, and the results in Table 1 mainly reproduce Harma et al. (2025). Sparsity rates only include 10 and 25%.

**Questions:**

1. What is the interplay between quantization and sparsity for other compression schemes, like SparseGPT, GPTQ, structured N:M sparsity?

---

> ### Author Response · Authors · 2025-11-21
>
> We greatly appreciate your review of the paper. We believe that the concerns and questions you raise are valid, and we will try to address them.
>
> Weaknesses:
> 1. We apologize for the limited novelty in the paper. However, we would like to briefly clarify some things. The goal was not to empirically show that $\mathbf{S} \to \mathbf{Q}$ is preferred to $\mathbf{Q} \to \mathbf{S}$, as shown previously in the literature. Instead, the goal was to try to prove this statement mathematically. Theorem 1 is indeed an extension of a result in Harma et al., but we wouldn't say it is completely useless, as it allows us to directly prove Theorem 2. Theorem 2 is a model level proof that has not been done in the literature.
> 2. We agree that the assumptions for Theorems 2 and 3 are quite strong. However, these assumptions were necessary to develop the math to show that in some settings $\mathbf{S} \to \mathbf{Q}$ is superior. As shown in Section 6, it is not generally true that $\mathbf{S} \to \mathbf{Q}$ outperforms $\mathbf{Q} \to \mathbf{S}$, so some assumptions needed to be made. We agree that the assumption that $\Delta \mathbf{w}\_{\mathbf{Q} | \mathbf{S}} = \Delta \mathbf{w}\_{\mathbf{Q}}$ is quite restrictive as acknowledged in the paper, but we believe that it is not unreasonable to assume that they are rather close as supported by the data in Table 1.
> 3. We again apologize for the lack of contribution and insight. As we acknowledge in the paper, QAS is a rather simple scheme. The main goal of QAS though was to understand the main limitations of $\mathbf{Q} \to \mathbf{S}$ and show how a simple scheme could potentially resolve it. As we state in the beginning of the paper, we focused mainly on post-training quantization and sparsification, where re-training is not required.
> 4. We understand that the empirical evidence is limited. As the focus of the empirics was to provide a counterexample though, we stuck to a relatively simple setting that we agree may not be very practical. We agree that Table 1 is in line with prior work, and it is used as our baseline. However, Table 2 shows that we present methodology that can actually contradict Table 1 and prior work. It shows that $\mathbf{Q} \to \mathbf{S}$ may be better under the proposed strategy.
>
> Questions:
> 1. We are certainly curious about this question too. Previous literature provides some empirical data that considers some of these compression schemes. The goal of our paper though was to try to analyze these phenomenon mathematically. Admittedly, we will need to use a simpler setup to show the superiority of $\mathbf{S} \to \mathbf{Q}$. However, we also show that once the assumptions break down, then it is not totally obvious that $\mathbf{S} \to \mathbf{Q}$ is superior, which complicates the current understanding in the literature.
>
> Thank you again for your review!

---

> > ### Comment · Reviewer_Sg9T · 2025-11-23
> >
> > I thank the authors for the detailed response. I don't have any further questions or comments.

---

### Official Review · Reviewer_cNqy · 2025-11-01

**Soundness:** 3
**Presentation:** 3
**Contribution:** 2
**Rating:** 2
**Confidence:** 4

**Summary:**

The paper analyses the effects of quantization and sparsification of a model and the order in which they are applied. Through tensor-level and model-level analysis, they show that sparsification then quantization is more preferred than quantization then sparsification, especially when the methods for quantization and sparsification are chosen at random. They also propose a new Quantization Aware Sparsification through which they show that quantization, then sparsification, can perform better than the other way around. They show empirical evidence of their claims using the OPT 125M parameter model.

**Strengths:**

1. They show that S -> Q is better than Q -> S through tensor-level analysis
2. They present the novel model-level analysis of the interaction between quantization and sparsification
3. They propose a new Quantization Aware Sparsification method that prunes quantized models
4. They validate the claims made through theoretical analysis with empirical evidence.

**Weaknesses:**

1. The specific scenarios where Q -> S is strictly better than S -> Q are not specific theoretically; only the possibility is shown.
2. No comparison to other quantization or sparsification methods.
3. The choice of AWQ for quantization seems arbitrary, and no justification is provided
4. No ablation study of how the proposed method scales with model sizes, sparsity levels, and bit-widths is shown

**Questions:**

Please see the weaknesses and
1. It would be better to explicitly mention the metric (perplexity; lower is better) used to evaluate the models.
2. In line 168, if $w_i$ is pruned, shouldn't the error be $|w_i|$?

---

> ### Author Response · Authors · 2025-11-21
>
> We greatly appreciate your review of the paper. We believe that the concerns and questions you raise are valid, and we will try to address them.
>
> Weaknesses
> We agree that the mathematical assumptions behind Theorem 4 are weak and understand that it could be misleading. Hopefully, we can clarify this a bit. The motivation of presenting this theorem was to try and see if a scenario with $\mathbf{Q} \to \mathbf{S}$ outperforming $\mathbf{S} \to \mathbf{Q}$ existed. We acknowledge this is not equivalent to saying that one must exist, but by providing the possibility, it naturally leads us to the counterexample that we present in Section 6. As this was a counterexample to show existence, we chose a specific quantization method (AWQ) and sparsification method, and we did not compare with other quantization and sparsification methods.
>
> Questions
> 1. Yes, we apologize for not expressing this clearly. We did attempt to notate this at the top of the table with a down arrow next to PPL indicating that lower perplexity is better.
> 2. Yes, that is correct. That is a typo on our part. It should read $\left|\mathbf{w}\_i - \overline{\mathbf{S}}\left(\overline{\mathbf{Q}}\left(\mathbf{w}\right)\right)\_{i}\right| = \left|\mathbf{w}\_i - \overline{\mathbf{Q}}\left(\overline{\mathbf{S}}\left(\mathbf{w}\right)\right)\_{i}\right|$.
>
> Thank you once again for your review!

---

### Meta-Review · Area_Chair_b486 · 2026-01-21

**Summary:**

The paper discusses the role of tensor sparsification and quantization. Specifically, the paper discusses whether sparsification or quantization should be applied first to lower the performance loss (measured by Lp norm).

All reviewers had major concerns about the novelty of the theoretical results and the assumptions used. For example, some key theoretical results are simple extensions of results in Harma et al. (2025), and the assumption that the Hessian is an identity matrix seems to be too strong.

Empirical evaluation also seems to be limited, with only one model setting and a few sparsity levels.

**Reviewer Concerns:**

While the rebuttal clarified a few confusions raised by the reviewers, the two major concerns (about the theoretical results and empirical evaluations) are not directly addressed by the authors.

Therefore, I recommend rejection of the paper. I encourage the authors to incorporate the reviewers' concerns in the next version of the paper.

**Reviewer Scores:**

All reviewers will likely maintain their scores, as their main concern about the lack of novelty of the main theoretical results is not resolved by the authors' rebuttal.

---

### Decision · Program_Chairs · 2026-01-26

Reject